# Cell swelling enhances ligand-driven β-adrenergic signaling

Alexei Sirbu [1], Marc Bathe-Peters [1], Jothi L. M. Kumar[2], Asuka Inoue [3,4], Martin J. Lohse[1,5,6] & Paolo Annibale [1,2] ✉

G protein-coupled receptors' conformational landscape can be affected by their local, microscopic interactions within the cell plasma membrane. We employ here a pleiotropic stimulus, namely osmotic swelling, to alter the cortical environment within intact cells and monitor the response in terms of receptor function and downstream signaling. We observe that in osmotically swollen cells the β2-adrenergic receptor, a prototypical GPCR, favors an active conformation, resulting in cAMP transient responses to adrenergic stimulation that have increased amplitude. The results are validated in primary cell types such as adult cardiomyocytes, a model system where swelling occurs upon ischemia-reperfusion injury. Our results suggest that receptors' function is finely modulated by their biophysical context, and specifically that osmotic swelling acts as a potentiator of downstream signaling, not only for the β2-adrenergic receptor, but also for other receptors, hinting at a more general regulatory mechanism.

One of the tenets in G protein-coupled receptors (GPCR) pharmacology is the ternary complex model, stating that a high-affinity state characterized by increased receptor-G protein coupling also correlates to increased affinity of the receptor towards its agonist, since the intracellular G protein coupling stabilizes the active receptor conformation[1]. Further to the development of this model, our understanding of GPCRs' function has evolved from two-state switches to rheostats exploring a complex conformational landscape, that in turn entails a broad repertoire of intracellular interactions[2]. Prototypical examples are molecules from the extracellular space acting as allosteric modulators of receptor function: they favor specific conformations -further stabilized upon ligand binding- that are different from those that would be observed in the absence of the modulator[3].

An emerging concept is that changes of the biophysical context around the receptor can 'allosterically' modulate its function. Mechanistically this involves the action of accessory proteins[4], local lipid composition[5], interaction with intracellular scaffolds[6], and -as highlighted recently- local curvature[7]. These interactions can modulate each of the canonical signaling steps (ligand binding, ternary complex

formation with cognate G protein, β-arrestin recruitment, trafficking), endowing these receptors with a broad modulatory repertoire able to nuance specific signals in a dynamic and flexible way in space and time.

Some of the interactions may be local, receptor-specific, and associated with specific modulatory proteins, such as Receptor Associated Modifying Proteins (RAMPs)[4]. Other interactions may be more spatially diffuse, such as the interaction of GPCRs containing a PDZ binding sequence with the cortical actin, or, as it was proposed more recently, the segregation of selected GPCRs to areas of higher or shallower curvature within the cell membrane[8]. Along these lines, we observed recently μm-sized receptor-specific segregation to a subcellular compartment, such as the T-tubular network of adult cardiomyocytes, characterized by high local curvature[9].

While the hypothesis that subcellular receptors' localization and local biophysical interactions may indeed modulate -to a sizable extent- their behavior is being increasingly explored[10,11], nonetheless the actual molecular mechanisms underpinning such effects are harder to unravel. These may be ascribed to heterogeneities in lipid composition of the plasma membrane[12], such as caveolae, or in the

[1]Max Delbrück Center for Molecular Medicine in the Helmholtz Association (MDC), Berlin, Germany. [2]School of Physics and Astronomy, University of St Andrews, St Andrews, UK. [3]Graduate School of Pharmaceutical Sciences, Tohoku University, Sendai, Japan. [4]Graduate School of Pharmaceutical Sciences, Kyoto University, Kyoto, Japan. [5]ISAR Bioscience Institute, Munich-Planegg, Germany. [6]Leipzig University, Medical Faculty, Rudolf-Boehm-Institute of Pharmacology and Toxicology, Leipzig, Germany. ✉e-mail: pa53@st-andrews.ac.uk

local density of the cortical actin network and the variability of its interactions with the membrane[13]. Moreover, if modulation of GPCR signaling indeed occurs across multiple length scales at the plasma membrane, from nanometer to micrometer scale, its microscopic identification challenges current technologies[14].

At the receptor level, this modulation ultimately relies on the notion that GPCRs are mechanosensitive molecules, i.e. able to register local stresses and forces acting on their transmembrane domains as well as intra- and extracellular loops, resulting in conformational landscapes favoring specific downstream signaling outcomes[15]. Indeed, the effect of mechanical tension along the plasma membrane has been invoked as the cause for the ligand-independent activation of selected GPCRs, such as the bradykinin B2 receptor[16], the angiotensin II type 1 receptor[17] or the Histamine H1 receptor[18]. The latter work identified helix 8 as the essential structural motif for GPCR mechanosensitivity, beyond the more established force sensing mechanisms of adhesion GPCRs, that instead rely on autocatalytic cleavage of the N-terminal domain.

We reasoned that osmotic swelling, a process that occurs in several (patho-)physiological settings may offer the type of pleiotropic biophysical modulation that we are seeking to test such hypothesis. Cell swelling is associated to a complex rearrangement of the cell membrane environment, ranging from alteration of local geometry, F-actin remodulation and increases in cell surface originating from flattening of the membrane, not unlike ironing out wrinkles in a cloth[19,20]. We decided to focus on β-adrenergic receptors (β-ARs), both because of their prototypical nature (i.e. they couple to the stimulatory G protein and elicit an increase in cAMP upon Gαs interaction with the adenylyl cyclase (AC)) but also because they display a rich conformational landscape[21] and the role of cortical actin in connection to

their localization and signaling, in particular the β2-AR, is well documented[6,22].

Here we show that swollen cells yield sizably higher cAMP concentrations in response to β-ARs stimulation than non-swollen cells. To pinpoint at what step this occurs, we set out to explore the whole signaling cascade of the β2-AR. Each step of the cascade was investigated with a specific fluorescent spectroscopy method, ranging from TIRF to fluorescence polarization and FRET biosensing, that allowed us working in intact cells. We could determine that this effect does not arise from alterations at the level of AC catalytic rates, nor by alterations of receptors trafficking, but rather by a distinctive increase in receptor-G protein coupling, mirrored by an increased affinity of the agonist isoproterenol for the receptor. These findings demonstrate a previously unappreciated way of modulating GPCR conformation by biophysical factors, while also showing the effect of cell swelling on 2nd messenger production, which has the potential to be transferred to other GPCRs or transmembrane receptors acting in a broad range of physiological settings.

## Results

To gauge the effect of osmotic swelling on the downstream signaling cascade of the β-ARs we first employed FRET-based cAMP fluorescence biosensors[23]. The layout of our readout is schematically depicted in Fig. 1a. When measuring FRET in a single cell under a microscope, it is possible to display the degree of energy transfer between the donor and acceptor of the sensor in a false-color scale, that will change upon agonist addition (100 pM isoproterenol (Iso) in our case). The relative change between the swollen and control state of a representative HEK293 cell is displayed in Fig. 1b, next to the donor and acceptor channels separately. Upon addition of the swelling medium

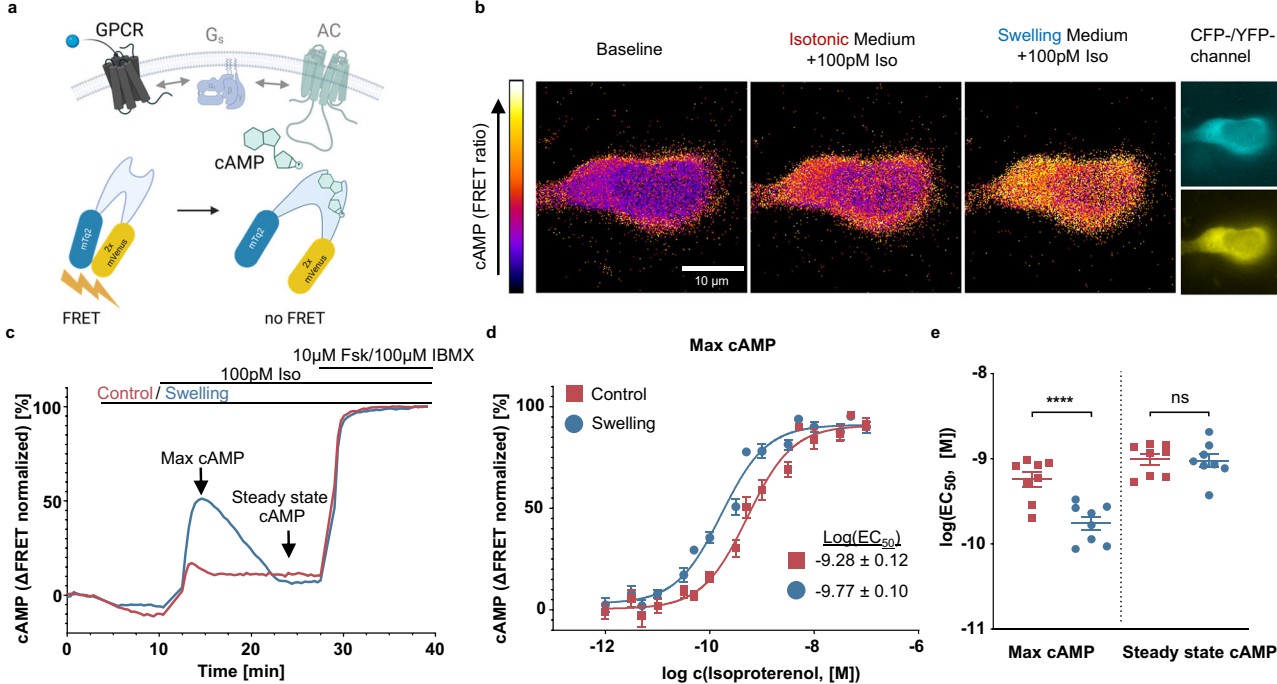

**Fig. 1 | Cell swelling increases β-adrenergic-mediated cAMP production in HEK293 cells. a** Schematics of GPCR-mediated cAMP production and its detection by a FRET biosensor (panel a created with BioRender.com released under a Creative Commons Attribution-NonCommercial-NoDerivs 4.0 International license: https://creativecommons.org/licenses/by-nc-nd/4.0/deed.en); **b** representative (out of 3 independent experiments) images of a HEK293 cell stably expressing cytosolic Epac-S^H187 [23] showing acceptor/donor ratio in false color; the cells are exposed to 100 pM isoproterenol first in isotonic (300 mOsm) and subsequently in swelling medium (200 mOsm); **c** representative curve (out of 8 independent experiments) showing kinetics of acceptor/donor ratio measured in HEK293 cells stably expressing Epac-S^H187 in a microplate reader (normalized to baseline and 10 μM forskolin + 100 μM IBMX); **d** averaged concentration response curves representing maximal cAMP concentrations, as indicated in (**c**) (mean ± SEM; n = 8 plates form 7 independent experiments); **e** log(EC50) values from individual experiments comparing swelling and control condition for maximal and steady state cAMP concentrations (mean ± SEM; n = 8 plates form 7 independent experiments; paired two-tailed *t*-test; *p* values: <0.0001 (Max cAMP), 0.7827 (Steady state cAMP). Source data are provided as a Source Data file.

(containing the same concentration of Iso as the isotonic medium) to a cell expressing endogenous levels of receptors, the cAMP response is sizably increased. We then moved to a high-throughput configuration where the signal from tens of thousands of cells was observed in 96-well plates using a fluorescent plate reader. A representative trajectory of intracellular cAMP concentration (proportional to the change in normalized FRET ratio) upon Iso addition in control vs swollen cells is displayed in Fig. 1c. Trajectories from swollen cells are displayed in blue throughout the manuscript, as opposed to cells maintained in isotonic medium (control) which are displayed in red. A strong peak in cAMP concentration is observed in the swollen cells a few minutes upon non-saturating Iso addition. Saturating cAMP levels are then obtained upon Fsk/IBMX stimulation for normalization purposes. We shall note that the results observed are independent of normalization to Fsk/IBMX (Supplementary Fig. 1a), that the raw data display clear antiparallel traces for donor and acceptor, both in control as well as in swelling conditions (Supplementary Fig. 1b) and that the response to FSK/IBMX is not sensitive to osmotic swelling (Supplementary Fig. 1c). When repeated for increasing concentrations of the agonist, ranging from 1 pM up to 100 nM concentration, it was thus possible to build a concentration response curve, as displayed in Fig. 1d. We observe a statistically significant difference in the $\log(EC_{50})$, which drops from $-9.28 \pm 0.12$ to $-9.8 \pm 0.1$ for the average curves. The peak cAMP response in swollen cells then declines within a few minutes to a steady state comparable to the concentration observed in the controls (Supplementary Fig. 1d). To rule out any effect of osmotic swelling on the readout of our fluorescence biosensor, a concentration response curve was calculated in the two conditions using the cell permeable cAMP derivative 8-Br-cAMP, displaying overlapping curves for swollen and control cells (Supplementary Fig. 1e). Individual $\log(EC_{50})$ values from separate experiments are displayed for the two conditions in Fig. 1e, which in turn allows to appreciate the fact that the observed difference applies only to the transient cAMP response and not to the steady state concentration. We verified that our time-resolved results using FRET biosensors are matched by an endpoint assay (10 min after stimulation), such as the AlphaScreen that is based on the readout of energy transfer between donor and acceptor beads, connected by biotinylated cAMP which is then displaced by the cAMP from cell lysates (Supplementary Fig. 1f).

Interestingly, similar results can be obtained when looking at cells stimulated with partial β-adrenergic agonists, such as salbutamol, salmeterol and terbutaline (Supplementary Fig. 2a–c). The specificity of the effects for both β1-AR and β2-AR was tested by overexpressing them separately in a cellular background devoid of endogenous β-AR expression, such as Chinese Hamster Ovary cells (CHO), as displayed in Supplementary Fig. 2d–e. When challenging other $G_s$ coupled receptors endogenously found in HEK293 cells, such as the histamine $H_2$ receptor or the melanocortin receptors (Supplementary Fig. 2 f–g) a similar outcome in terms of cAMP response can be observed upon osmotic swelling. Moreover, we observed a similar behavior at the level of Gi protein activation[24] for the μ opioid receptor (μOR) upon stimulation with the synthetic agonist DAMGO (Supplementary Fig. 3a–c).

We then addressed the question if this remarkable behavior was observed also in primary cells that are known to be subject to osmotic swelling in physiological or pathophysiological conditions, namely adult ventricular cardiomyocytes (CM): CMs undergo swelling during the reperfusion phase following ischemic shock[25]. Murine adult ventricular CMs were isolated and seeded on multiwell imaging plates and imaged on the same day, after labeling using a cell membrane dye (Materials and Methods). Upon exposure to the swelling media the CMs display a visible increase in volume, as illustrated in Fig. 2a. The increase was quantified as the relative change in cell area, as detected in 2D confocal sections (Fig. 2b, Supplementary Fig. 4a, b). The volume of the cells appears to increase for approximately 10-15 minutes, before saturating. This is in line with what is observed for HEK293 cells,

where saturation is reached after ~15 min from the onset of osmotic swelling (Supplementary Fig. 4g, h). Following the same approach used to generate the data in Fig. 1b, we employed CMs isolated from a transgenic mouse expressing the Epac1-camps cAMP sensor[10] and imaged them under a microscope (Supplementary Movie 1). The relative changes in cAMP concentration upon agonist addition were quantified for each cell, and representative traces for a control vs a swollen CMs are displayed in Fig. 2c. Upon osmotic swelling, the cAMP response to non-saturating concentrations of Iso was clearly enhanced, consistent with the observations reported in Fig. 1 for HEK293 cells.

The data for 1 nM and 10 μM iso additions are summarized in Fig. 2d and illustrate that cAMP signaling is enhanced in swollen adult CMs for Iso stimulations around the sub-maximal concentration[17], while being equal at saturating concentration. Albeit it was not possible to generate a full concentration response curve based on single cell images, this data reports an effect clearly consistent with the one displayed in Fig. 1d. In order to perform FRET experiments, contractility in adult CMs was inhibited. However, we tested the effect of osmotic swelling on on the chronotropic response of human induced pluripotent stem cells-derived cardiomyocytes (hiPSC-CMs). Here, we could show that, upon iso stimulation (100 nM) (Supplementary Fig. 5a) the beating rate in swollen cells was significantly increased over cells in isosomotic conditions (Supplementary Fig. 5b, c). Overall, these results point to the potential (patho-)physiological relevance of this mechanism.

We thus set out to identify the molecular determinants of the observed changes in cAMP response in swollen cells. Since swelling is a pleiotropic modulator of cell homeostasis, we systematically investigated all steps of the signaling cascade, from PDE-mediated degradation of cAMP up to G protein coupling. We graphically summarized these steps in Fig. 3a (1-6), that acts as a legend to the other panels of Fig. 3. In Fig. 1 we have reported on (1), namely the observed increase in cytosolic cAMP. As our next step we measured cAMP concentrations with a biosensor localized to the plasma membrane (Epac1-camps-CAAX)[24] to test if the observed cytosolic increase in cAMP can be ascribed to changes in local membrane pools (2). This explanation could be excluded after observing an analogous pattern in local cAMP concentrations at the cell membrane (Supplementary Fig. 6a). We then set to rule out that altered receptor trafficking (3), namely decreased desensitization or downregulation supported by clathrin-mediated endocytosis could be the origin of the observed effect[26]. First, we monitored whether baseline receptor expression was altered at the plasma membrane of swollen vs control cells, observing that the average number of heterologously expressed β2-AR is not affected by the treatment (Supplementary Fig. 6h–i). We then treated cells with Dyngo-4a, a well characterized dynamin inhibitor: supplementary Fig. 6b illustrates that the cAMP production increase reported in Fig. 1 is still present in HEK293 cells treated with 10 μM Dyngo-4a. Moreover, the decrease in the number of receptors at the plasma membrane in cells where internalization takes places is comparable between control and swollen cells after 10 minutes of Iso stimulation, confirming that swelling has limited to no effect on receptor trafficking in the observed timeframe (Supplementary Fig. 6j). We then investigated the potential role of altered phosphodiesterase activity[27] (4), as altered degradation rates (in this case reduced) may account for the observed increase in cAMP concentration transients in swollen cells.

The addition of the pan-PDE inhibitor IBMX, at the concentration of 100 μM, did not abolish the observed relative increase in transient cAMP concentration in the swollen cells, as illustrated in Fig. 3b and Supplementary Fig. 6c[27]. Cholesterol sequestration (5) using methyl-β-cyclodextrin (MβCD) did not abolish the observed effect either (Supplementary Fig. 6d). We thus moved upstream along the signaling cascade: the next target is the family of enzymes responsible for cAMP synthesis, namely ACs (6). ACs are canonically activated by the GTP-

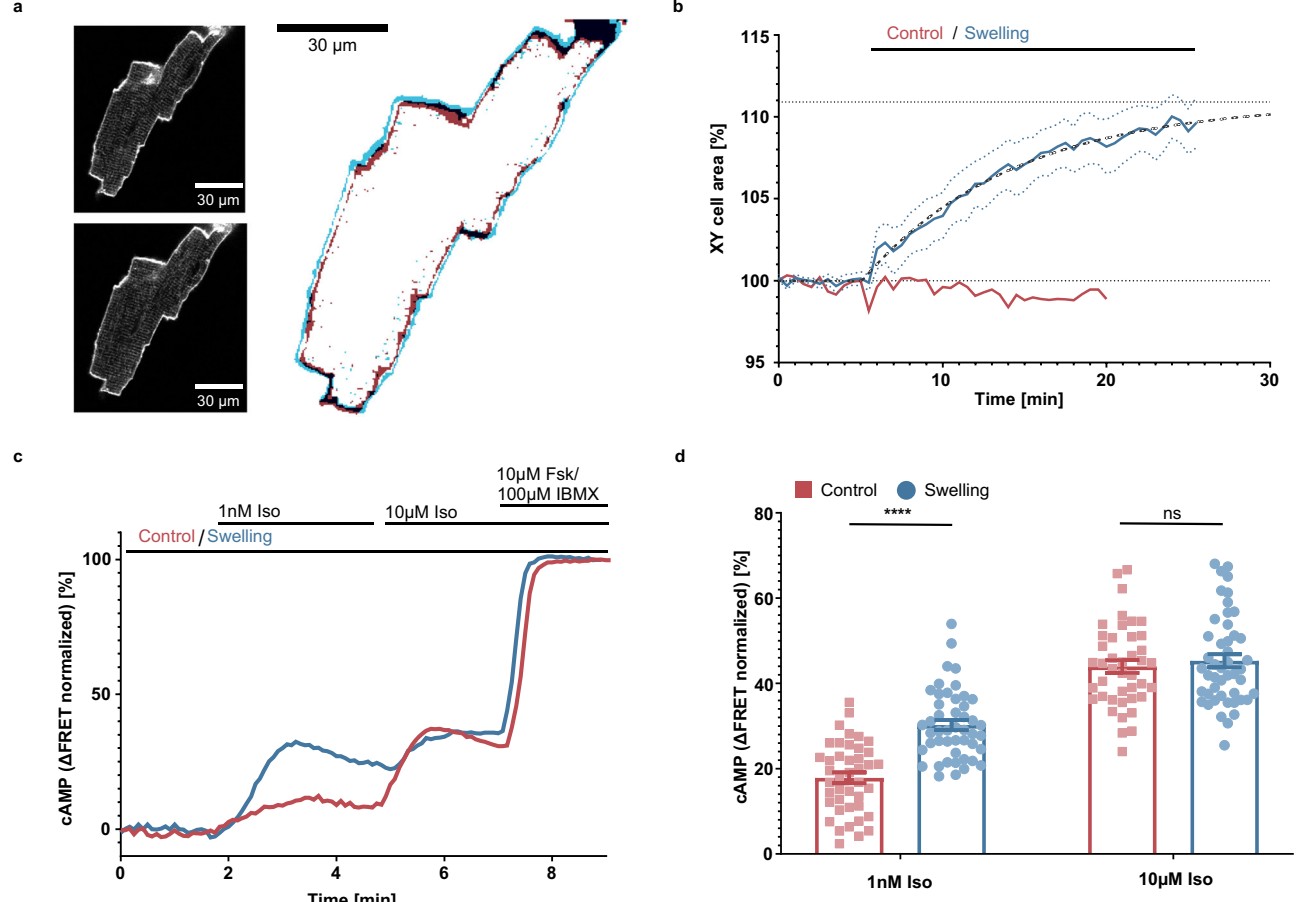

**Fig. 2 | Cell swelling increases β-adrenergic mediated cAMP production in adult mouse ventricular cardiomyocytes (CM). a** Representative confocal images (out of 6 cells from 3 independent experiments) of CM under isotonic conditions (upper inset) and after 20 minutes of exposure to swelling medium (lower inset), with an overlay of cell edges under both conditions (main); **b** average curve showing the area of a confocal CM slice (mean ± SEM; $n = 6$ cells from 3 independent experiments); **c** representative curve (out of 3 independent experiments) showing kinetics of acceptor/donor ratio measured in CM stably expressing Epac1-camps under an epifluorescence microscope (normalized to baseline and 10 μM forskolin + 100 μM IBMX); **d** maximal cAMP concentrations reached upon stimulation with 1 nM and 10 μM Iso respectively, normalized to 10 μM forskolin + 100 μM IBMX (mean ± SEM; $n = 42$ (control) and 47 (swelling) cells from 3 independent experiments; unpaired two-tailed $t$-test; $p$ values: <0.0001 (1 nM Iso), 0.5359 (10 μM Iso)). Source data are provided as a Source Data file.

bound $G\alpha_s$ subunit of the heterotrimeric G protein. The diterpene forskolin and $G\alpha_s$ have a synergistic/cooperative action in activating ACs[28–30]. In cells possessing endogenous $G\alpha_s$, direct stimulation of AC by forskolin leads to marked cAMP concentration increases, and the differential effect elicited by cell osmotic swelling persists (Fig. 3c and Supplementary Fig. 6e). However, interestingly, in cell lines carrying a $G\alpha_s$ knockout[31], we observed that any effect of osmotic swelling upon the amplitude of the cAMP concentration increase (Fig. 3d and Supplementary Fig. 6f) is lost. The effect is however recovered upon heterologous transfection of $G\alpha_s$ in the knockout lines (Supplementary Fig. 6g), thus pinning on the α subunit of the heterotrimeric $G_s$ protein a key role in modulating the signaling cascade's response to osmotic swelling. The overall results are summarised as $\log(EC_{50})$ values in Fig. 3e, that shows a consistent left-shift of the $EC_{50}$ values of approximately half a log unit.

Based on these observations we decided to investigate further the role of $G_s$ in mediating this remarkable response of cAMP transients to isoproterenol stimulation upon cell swelling. To do so we first asked the question if $G_s$ activation by the receptor is enhanced in swollen cells. This question was addressed exploiting fluorescently tagged nanobody 37 (Nb37)[32], reported to bind the active guanine-nucleotide-free $G\alpha_s$ found in the β2-AR - $G_s$ ternary complex, as schematically illustrated in Fig. 4a. After co-transfecting HEK293AD cells with β2-AR, the heterotrimeric G protein and Nb37-eYFP (as discussed in Materials

and Methods) we imaged the cell basolateral membrane using Total Internal Reflection Fluorescence[33], to enhance the signal from membrane-associated Nb37-eYFP. This approach allowed us to monitor the membrane recruitment of Nb37-eYFP upon increasing concentrations of Iso (Fig. 4b and Supplementary movie 2), and in turn confirm that in swollen cells the recruitment is enhanced at non-saturating isoproterenol concentrations (Fig. 4c). Overall, the relative recruitment of Nb37 is almost double in swollen cells than in the control (Fig. 4d). This suggests that in swollen cells there is a sizably higher proportion of active $G_s$, hinting at formation of a higher number of Iso - β2-AR - Gs ternary complexes, and therefore that osmotic swelling is a positive modulator of the receptor active conformation. These measurements are mirrored by measuring β-Arrestin-2 recruitment (Fig. 4e, f), that is also enhanced in swollen cells (Fig. 4g). We observed an almost twofold increase of eYFP-tagged β-arrestin-2 (β-Arr-2-eYFP) at the basal cell membrane upon non-saturating isoproterenol stimulation (Fig. 4h). Interestingly, β-Arrestin-2 recruitment in swollen $G\alpha_s$ knockout cells was more prominent than in non-swollen cells (Supplementary Fig. 7), suggesting that the effects observed at the arrestin are independent and not a downstream effect of $G\alpha_s$ activation. It is important to note, that fluorescence intensity of a membrane anchored fluorescent protein (Venus-CAAX) was not significantly affected by cell swelling, indicating no effect of cell morphology on the TIRF measurements (Supplementary Fig. 8).

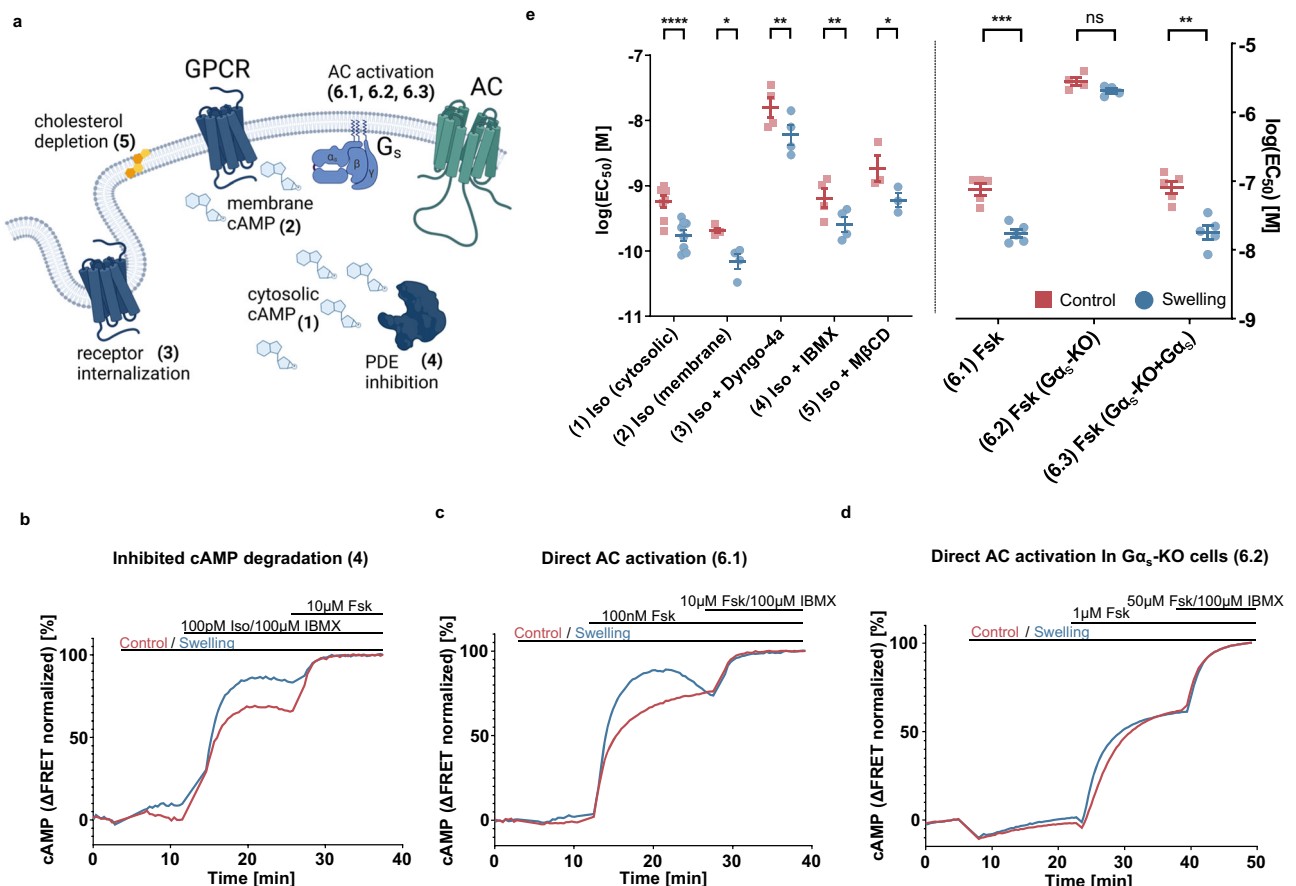

**Fig. 3 | Impact of cell swelling on the molecular steps of the β-adrenergic signaling cascade. a** Schematics of key steps of the signaling cascade that may be affected by swelling and were assessed under following conditions: (1) cytosolic cAMP upon Iso stimulation, (2) local cAMP at the plasma membrane upon Iso stimulation measured with an Epac1-Camps-CAAX sensor, (3) cytosolic cAMP measured upon Iso stimulation after treatment with 10 μM Dyngo-4a for 30 min, (4) cytosolic cAMP measured upon Iso stimulation with simultaneous addition of 100 μM IBMX, (5) cytosolic cAMP measured upon Iso stimulation under cholesterol depletion after incubation for 30 minutes with 10 mM methyl-β-cyclodextrine, (6) cytosolic cAMP measured upon Fsk stimulation in wt Gαs phenotype (6.1), Gαs-KO cells (6.2), and Gαs-KO cells transiently transfected with Gαs (6.3) (panel **a** created with BioRender.com released under a Creative Commons Attribution-NonCommercial-NoDerivs 4.0 International license: https://creativecommons.org/licenses/by-nc-nd/4.0/deed.en); **b**–**d** representative kinetic cAMP concentration curves (*n* numbers listed below) obtained under respective conditions described in **a**; **e** log(EC50) values from individual experiments comparing swelling and control condition for maximal cAMP concentrations under conditions listed in **a** (mean ± SEM, $n(1) = 8$, $n(2) = 4$, $n(3) = 4$, $n(4) = 4$, $n(5) = 3$, $n(6.1) = 5$, $n(6.2) = 4$, $n(6.3) = 5$ independent experiments; paired two-tailed *t*-test; *p* values: <0.0001 (1), 0.0212 (2), 0.0026 (3), 0.0022 (4), 0.0478 (5), 0.0002 (6.1), 0.081 (6.2), and 0.0036 (6.3)). Source data are provided as a Source Data file.

If osmotic swelling indeed favors the formation of a ternary agonist-receptor-$G_s$ (and/or agonist-receptor-β-arrestin-2) complex, we expect this to be reflected in a direct modulation of receptor conformation[34]. To test for this possibility, we conducted a set of assays aimed at monitoring agonist binding and receptor activation in intact cells. We employed a recently reported fluorescent ligand against the β-ARs, based on the inverse agonist carazolol[9], to conduct a set of fluorescence anisotropy binding assays[35], as graphically illustrated in Fig. 5a. Simply put, as the the fluorescent ligand binds, its fluorescent anisotropy increases as a result of the reduction in rotational diffusion. As the fluorescence anisotropy increases proportionally to the fraction of bound ligand, it is possible to incubate cells overexpressing β2-AR as well as heterotrimeric $G_s$ and Nb37 with a fixed concentration of fluorescent ligand, and then measure changes in fluorescence anisotropy as the fluorescent ligand is progressively displaced by increasing concentrations of the non-fluorescent Iso (Fig. 5b). When comparing swollen and control cells, this leads to a shift in the ligand $K_i$, from $\log(K_i) = -6.77 \pm 0.03$ to $-6.98 \pm 0.04$ (Fig. 5b). Albeit the observed shift is small, it is significant, and it was consistently observed for each plate we investigated, as graphically summarised by the paired comparisons displayed in Fig. 5c.

We can probe receptor activation from the intracellular side by using a fluorescent probe acting as a proxy for the intracellular proteins recognizing the active receptor conformation, namely fluorescently tagged Nb80 (Nb80-eYFP)[36], as schematically depicted in Fig. 5d. By employing TIRF to measure nanobody recruitment, we can observe that there is a significant enhancement in Nb80 recruitment in swollen cells (Fig. 5e), as also reflected by the full concentration response curve's ($\log(EC_{50})$) shifts from $-7.96 \pm 0.06$ to $-8.28 \pm 0.09$ (Fig. 5g).

These results appear to be confirmed by conducting the experiments presented in Fig. 5 in cells preincubated with saturating (100 μM) concentration of the β-AR antagonist propranolol. In these conditions, no cAMP elevation is observed in control cells upon either 1 μM or 10 μM, but the blockade appears to be abolished in swollen cells, where cAMP response can be observed already when 1 μM isoproterenol is added to the cells (Supplementary Fig. 9a, b). Interestingly, in swollen cells, a partial breakage of the propranolol blockade can be observed already at isoproterenol concentrations as low as 10 nM (Supplementary Fig. 9c, d). These observations support the notion that cell swelling transiently promotes an active receptor conformation, that favors agonist binding.

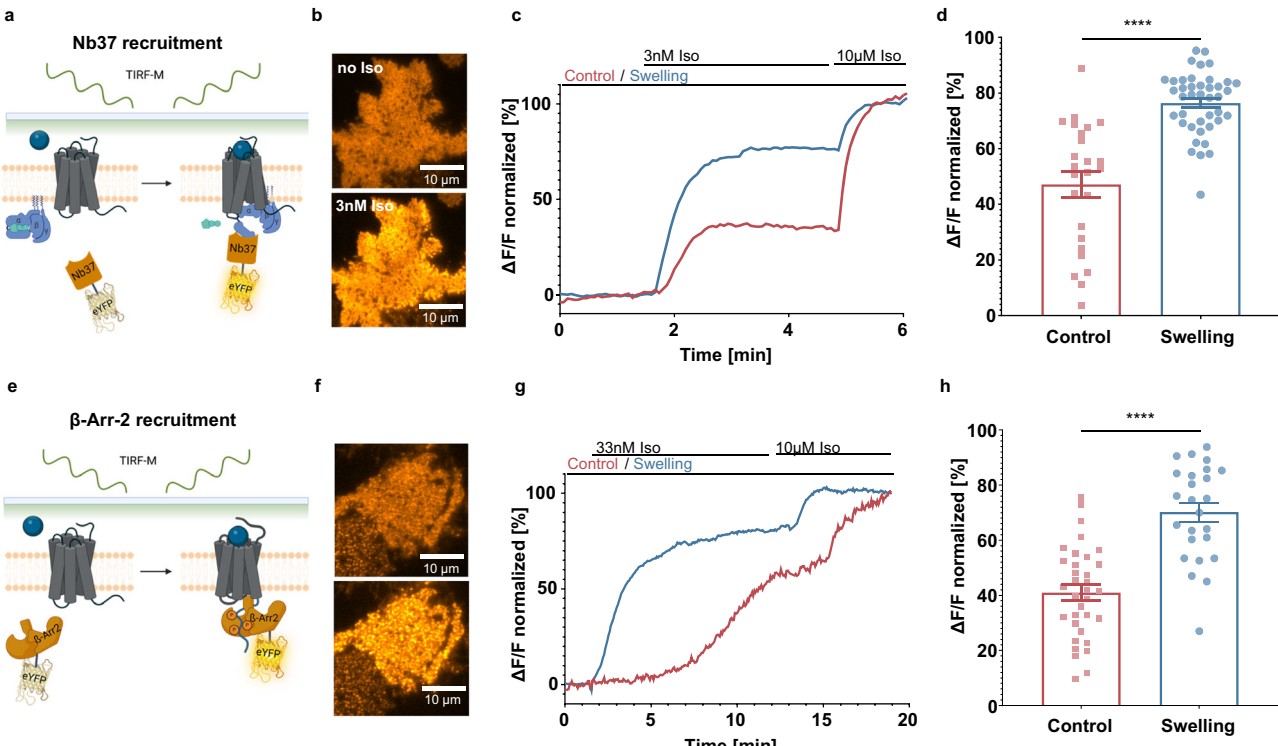

**Fig. 4 | Swelling leads to an increased effector recruitment. a** $G_s$ activation by the β2-AR is monitored by recruitment of the fluorescently tagged nanobody 37 (Nb37) at the plasma membrane of the cell together with overexpression of heterotrimeric $G_s$ (panel a created with BioRender.com released under a Creative Commons Attribution-NonCommercial-NoDerivs 4.0 International license: https://creativecommons.org/licenses/by-nc-nd/4.0/deed.en); **b** representative (of $n = 25$ (control) and 41 (swelling) cells from 3 independent experiments) TIRF-M images of Nb37-eYFP expressed in HEK293-AD cells before and after application of 3 nM isoproterenol; **c** representative curve showing relative increase of membrane fluorescence collected as first 3 nM, and then at saturating 10 μM concentration of isoproterenol is added to the cell (normalized to baseline and 10 μM Iso response); **d** relative fluorescence increase measured upon Nb37-eYFP recruitment upon 3 nM isoproterenol stimulation in swollen vs non-swollen cells (mean ± SEM; $n = 25$ (control) and 41 (swelling) cells from 3 independent experiments; unpaired two-

tailed $t$-test, $p$ value < 0.0001); **e** recruitment of fluorescently tagged β-Arrestin-2 (β-Arr-2) by the β2-AR is monitored by its recruitment to the plasma membrane of the cell (panel e created with BioRender.com released under a Creative Commons Attribution-NonCommercial-NoDerivs 4.0 International license); **f** representative (of $n = 34$ (control) and 25 (swelling) cells from 3 independent experiments) TIRF-M images of βArr-2-eYFP expressed in HEK293-AD cells before and after application of 33 nM isoproterenol; **g** representative curve showing relative increase of membrane fluorescence collected as first 33 nM, and then at saturating 10 μM concentration of isoproterenol is added to the cell (normalized to baseline and 10 μM Iso response); **h** relative fluorescence increase measured upon β-Arr-2-eYFP recruitment upon 33 nM isoproterenol stimulation in swollen vs non-swollen cells (mean ± SEM; $n = 34$ (control) and 25 (swelling) cells from 3 independent experiments; unpaired two-tailed $t$-test; $p$ value < 0.0001). Source data are provided as a Source Data file.

Altogether these data confirm that osmotic swelling modulates the β2-AR conformational landscape, favouring an active conformation that in turn leads to an enhanced downstream cAMP cascade.

## Discussion

Our current understanding of the GPCR signaling cascade, from the ternary complex model down to the more recent structural determination of receptor conformational states, is facing a rapid and sustained evolution. Cartoons displaying GPCR signaling in textbooks still typically highlight only the receptor at the membrane, its ligand, and the downstream interaction partner, either the heterotrimeric G protein or, more frequently now, β-arrestins. However, literature reports indicate that GPCRs are exposed to many specific intracellular interactions with other proteins as well as to the modulatory effects of other, less described, interactions.

In particular, the intracellular domain of plasma membrane GPCRs is exposed to the cell cortex and its distinct organization[37]. In this work we decided to investigate a more general mechanism that can fundamentally alter the cortex and at the same time has a role in several (patho-)physiological processes, ranging from cardiac ischemia reperfusion[25] to renal function[38], namely osmotic swelling. In our hands, osmotic swelling became a tool to induce pleiotropic changes of the cortex and cell membrane, thereby affecting receptor function.

While osmotic swelling has been used before in the context of GPCR signaling[16,18], in these instances the GPCRs are observed to undergo activation upon pure mechanical stimulation. This is fundamentally different from our observations, that do not display spontaneous activation upon hypotonic shock at 200 mOsm, but only an effect upon additional agonist stimulation. Interestingly, the only report that discusses the ligand-dependent stimulation of a GPCR in cells undergoing osmotic swelling discusses the allosteric effect of hypotonic shock in favoring ligand-dependent recruitment of β-arrestin 2 to the AT1R[39], an effect we now also see for the β-ARs.

It shall be noted that we explore receptor function in the few minutes upon the onset of osmotic swelling, thus restricting the number of potential effects impacting the signaling cascade to rapid responses, arising from a local reorganization of the membrane and cortex (Supplementary Fig. 10), while excluding more lasting effects such as recruitment of phospholipids to the plasma membrane[40] or alterations to gene expression patterns[41]. On this timescale the overall curvature radius of the membrane decreases as the volume increases, while at the same time, small-scale ruffles on the plasma membrane, including caveolae, are 'flattened out' together with a remodulation of the actin cortex[40]. Caveolae are indeed recognized as important mediators of membrane mechanical response to external stresses[42] and their role in adrenergic signaling is also well established, especially

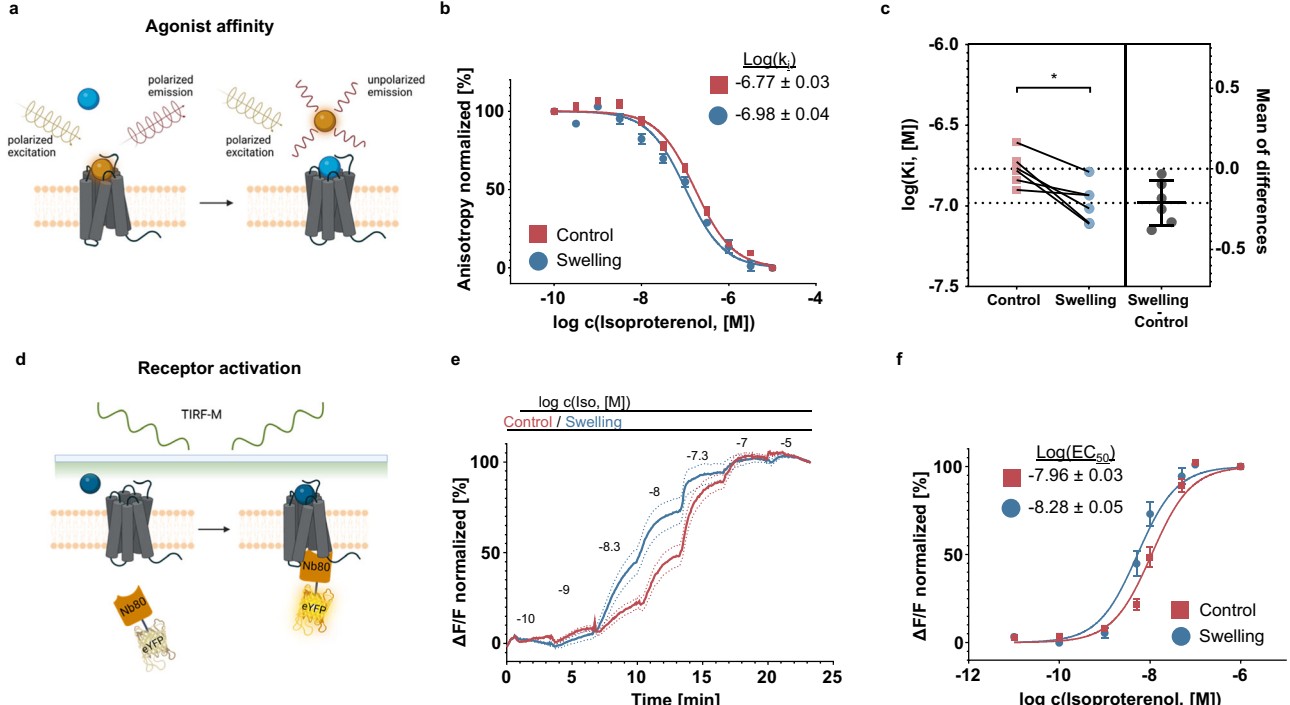

**Fig. 5 | Swelling favors an active receptor conformation. a** Ligand binding affinity is determined by employing a fluorescently labeled ligand and measuring the degree of fluorescence anisotropy (panel **a** created with BioRender.com released under a Creative Commons Attribution-NonCommercial-NoDerivs 4.0 International license: https://creativecommons.org/licenses/by-nc-nd/4.0/deed.en); **b** average curve of increasing concentrations of isoproterenol displacing 2 nM of fluorescently labeled JE1319 from the β2-AR in swollen and non-swollen cells with the corresponding log(Ki) values; **c** pairwise comparison of log(Ki) values obtained from single experiments and the average of the difference overlay (for **b** and **c** - mean ± SEM; $n = 6$ plates from 5 independent experiments; paired two-tailed $t$-test; $p$ value 0.012); **d** conformational activation of the β2-AR is monitored by recruitment at the plasma membrane of the cell by employing the fluorescently tagged nanobody 80 (Nb80) (panel **d** created with Biorender.com released under a Creative Commons Attribution-NonCommercial-NoDerivs 4.0 International license: https://creativecommons.org/licenses/by-nc-nd/4.0/deed.en); **e** averaged time-sequence of the relative increase of membrane fluorescence collected as increasing concentrations of Iso are added to the cells (normalized to baseline and 10 μM Iso response); **f** the resulting averaged concentration response curves for cells in isotonic and swelling media and associated log(EC50) values (for **e** and **f** - mean ± SEM; $n = 46$ (control) and 45 (swelling) cells from 5 independent experiments). Source data are provided as a Source Data file.

in relation to β2-AR signaling[43]. Rasenick and colleagues reported that caveolae (disrupted by hyposmotic swelling) attenuate Gs signaling in response to isoproterenol. However, in their experiments, methyl-β-cyclodextrin (MβCD) treatment did not alter basal cAMP levels[44]. Osmotic swelling was further reported to specifically disrupt Caveolin1 – Gq protein interaction and attenuate Gq-mediated Ca²⁺ release[45]. In this work, the same effect on Gq signaling is observed upon cholesterol depletion under MβCD treatment, which leads to perturbation in caveolae structure. Interestingly, the same treatment was reported to increase beta-adrenergic mediated response to isoproterenol[46]. This would be in line with the observation that cell swelling leads to a reduction in number of caveolae[42]. Overall, there is thus a large and often conflicting mole of data on the effects of long-term/permanent modifications of membrane microdomains on GPCR signaling.

On the backdrop of this scenario, we observe a significant increase in cAMP response in cells undergoing osmotic swelling, unaffected by MβCD disruption (Supplementary Fig. 6d). None of the previously mentioned works addressed the effects on cAMP caused by mild hypoosmotic swelling, which in our observations are transient. Moreover, our results show that a similar swelling-induced response is observed, besides the β-AR system, also for receptors that are not traditionally associated to caveolae, such as histamine receptors or melanocortin receptors (Supplementary Fig. 1F–H).

We thus set out to dissect all steps of the β-AR signaling pathway in intact cells, to pinpoint where swelling is having an impact on the signaling cascade. In the course of our investigation, we adapted and implemented a wide set of fluorescence spectroscopy techniques that allowed monitoring the activity of each of the signaling partners that support cAMP production and its regulation in single, living cells. Fluorescence cAMP biosensing allowed us to report on the effect (Fig. 1) in HEK293 cells as well as in the more physiological setting of adult cardiomyocytes (Fig. 2). The physiological implication of the enhanced cAMP production was observed in the form of an increased beating rate of hiPSC-CM cells in response to isoproterenol (Supplementary Fig. 5).

This was then combined with specific drug combinations and knockout cell lines to rule out the role of trafficking (Supplementary Fig. 6b,h–j), cAMP degradation (Fig. 3b and Supplementary Fig. 6c) or an altered activity of the ACs (Fig. 3c,d and Supplementary Fig. 6e–g). Indeed, Fsk-induced AC activity appears unaffected by swelling in the absence of the Gs, suggesting that a key player in modulating the signaling cascade of the β-ARs is indeed the stimulatory heterotrimeric G protein. To further appreciate how Gs role in transducing receptor signaling is modulated upon osmotic swelling, we employed a fluorescent reporter for the active conformation of Gαs, namely eYFP tagged Nb37, a nanobody originally developed to stabilize the structure of the active β2-AR in complex with Gαs[47].

With this assay, we can then demonstrate that in swollen cells an equal concentration of agonist leads to an increased number of active receptor-Gαs complexes (Fig. 4a–c), as well as an enhanced recruitment of β-Arrestin-2 (Fig. 4d–g). These data may be explained either by the β-arrestin-2 wave following the successful recruitment and dissociation of Gs at an increased number of receptors in swollen cells, or by osmotic swelling promoting a receptor active conformation that is

non-selective of Gα$_s$ vs β-arrestin-2. Nonetheless, these results point to improved receptor to G protein coupling as the driving mechanism behind the observed increase in cAMP upon osmotic swelling. According to the ternary complex model[1], we thus investigated the possibility that osmotic swelling stabilizes the active receptor conformation first by monitoring ligand binding in intact cells (Fig. 5a) and then by a proxy for G protein coupling, namely the Nb80[36] (Fig. 5d).

Upon ligand displacement experiments, the measured k$_d$ displays a statistically significant change in swollen cells (Fig. 5b, c), with increased affinity of the receptors towards isoproterenol in swollen cells. Moreover, the enhancement in Nb80 recruitment upon swelling displayed in Fig. 5e, f corroborates the notion that a receptor active conformation is indeed favored by osmotic swelling.

Overall, these results point to a modulation of receptor activity upon cell swelling, that we ascribe to cell swelling favouring the active conformation of the β2-AR. This observation would agree with a recent report suggesting that osmotic swelling leads to increased hydration of the intracellular side of the receptor, 'forcing open' the G protein binding pocket and thus favouring coupling[48]. Other explanations, such as an enhanced 'collision coupling' of the receptor to the G protein[49], are possible: while we observed a slight increase in receptor diffusion coefficients, we did not measure any increase in G protein (Gα$_s$) diffusion rate in swollen cells as opposed to control (Supplementary Fig. 11). Since, as discussed, osmotic swelling has broad effects on the cell cortex and plasma membrane, such as actin remodeling, altered lipid composition, as well as changes in both local (flattening of ruffles and invaginations) and cell-wide (overall increase in volume) geometrical curvature[41], we believe it is difficult to pinpoint its effect to one specific molecular mechanism. However, the molecular mechanisms at play must directly scale with the osmolarity, as treatments with media of different osmolarity are observed to differentially affect cAMP concentrations, including a reduction of cAMP production by cell shrinkage (Supplementary Fig. 12). Nonetheless, our observation with other G$_s$-coupled receptors as well as with the G$_i$ coupled μOR, point to the generality of this regulatory mechanism.

These results suggest that osmotic cell volume changes can be intrinsic modulators of GPCR activity in several contexts of (patho-) physiological relevance, ranging from cardiac ischemia-reperfusion to the reported swelling of adipocytes[19]. In the heart such preference towards an active β2-AR state in swollen cells could potentially diminish the effect of antagonist administration. This is indeed what we observe: strikingly, in our measurements cell swelling leads to receptor activation by isoproterenol (as mimicked by Nb80 recruitment) even after an incubation with saturating concentration of propranolol, a prototypical β-blocker (Supplementary Fig. 9).

Furthermore, cAMP has also been reported to be actively involved in cell volume regulation. It has been shown to support regulatory volume decrease in swollen cells[50] by additive stimulation of volume-regulated Cl$^-$ efflux[51,52]. This may suggest an adaptive mechanism of swollen cells, that elevate normal agonist-dependent cAMP levels to favor return to normal volume.

Finally, membrane permeability for cAMP was shown to be increased upon osmotic swelling[53], potentially creating the need of increased production rate to compensate the efflux

Further studies and more refined biophysical approaches will be needed to deconvolve the specific molecular mechanisms mediating the effects of a pleiotropic perturbation such as osmotic swelling at the level of the isolated receptor. At the same time, additional functional investigations can shed light on the specific role of GPCR-mediated cAMP transient in the context of volume regulation and pathophysiological conditions like cardiac ischemia/reperfusion.

## Methods

Our research complies with all relevant ethical regulations. All animal experiments were carried out according to the German animal welfare act considering the guidelines of the National Institute of Health and the 2010/63/EU Directive of the European Parliament on the protection of animals used for scientific purposes. Animal experiments were conducted with the approval of the Landesamt für Gesundheit und Soziales (LAGeSo, Berlin) and the Max-Delbrück-Center for molecular medicine. hiPSC were commercially sourced (GM25256) and were not genetically edited, and as such are not considered relevant material under the 2004 UK Human Tissue Act and their use did not require specific ethical approval.

### Cell culture

HEK293 were HEK-tsA201 cells (Sigma-Aldrich ECACC Cat#96121229). HEK293AD cells were from Biocat, cat. no. AD-100-GVO-CB. Gα$_s$-KO cell line were derived from HEK293A from Thermo Fisher Scientific. HEK293 derived cells (Epac-S$^{HI87}$) were derived from HEK-tsA201. Cells were cultured in DMEM medium containing 4.5 g/l glucose (Gibco) supplemented with 10% fetal calf serum (FSC, Biochrome), 2 mM L-glutamine, 100 units/ml penicillin and 0.1 mg/ml streptomycin, at 37 °C, 5% CO$_2$. To split cells, growth medium was removed by aspiration and cells were washed once with 10 mL of PBS (Sigma), followed by trypsinization for 2 min in 1.5 mL of trypsin 0.05%/EDTA 0.02% (PAN Biotech) solution and resuspended in the desired amount of DMEM medium.

### Transfection

For the single cell microscopy experiments the cells were transfected directly in the imaging dishes 24 hours prior to conducting the experiments, using JetPrime transfection reagent (Polyplus) according to manufacturer's protocol. For the plate reader experiments the cells were transfected on 10 cm plates 48 hours before the experiment using a reduced amount of JetPrime transfection reagent and total DNA (5 μl and 2.5 μg respectively) and transferred to black 96-well plates (Brandt) 24 hours before the experiment. The transfected cDNA and cDNA ratios for co-transfections are listed in further sections.

### Molecular cloning

The tricistronic G$_s$ and G$_s$-CFP plasmids were created using NEBuilder HiFi DNA assembly cloning kit (NEB). Gα$_s$ (or Gα$_s$-CFP[54]), Gβ$_1$ and Gγ$_2$ were cloned into the pcDNA3.1 backbone following the same structure as for the published G$_i$ FRET sensors[24]: Gβ$_1$−2A-Gγ$_2$-IRES-Gα$_s$ (or Gα$_s$-CFP respectively). The primers used to generate these constructs are available in Supplementary Table 1.

### Swelling medium

Throughout all experiments two types of imaging buffers (control – mannitol electrolyte solution 300 mOsm (MES300) and swelling – mannitol electrolyte solution 200 mOsm (MES200)) were used. A basic buffer with a low osmolarity – ES150 – was produced by solving 60 mM NaCl, 5 mM KCl, 2 mM CaCl$_2$, 1 mM MgCl$_2$, 10 mM HEPES in distilled water and resulting in electrolyte solution 150 mOsm (ES150). It was then supplemented with 150 mM mannitol (Sigma) to produce MES300 and reach 300 mOsm. MES300 was then used for washing and incubation steps as described in specific experiments. To achieve hypotonic conditions of 200 mOsm during the experiments, 2/3 of the total volume of MES300 was removed from the wells and replaced with the same volume of ES150, resulting in MES200. In control condition 2/3 of MES300 were also removed and replaced with fresh MES300 to avoid ascribing any effects to mechanical artifacts.

### Cardiomyocyte isolation

Cardiomyocyte isolation involved obtaining cardiac myocytes from adult ventricles of seven (7) CAG-Epac1-camps mice aged 8 to 12 weeks[9], including 6 males and 1 female. The CAG-Epac1-camps mice (mus musculus, FVB/N transgenic mice) were a kind gift of Kristina Lorenz, Institute for Pharmacology and Toxicology, University of

Würzburg. All animal experiments were carried out according to the German animal welfare act considering the guidelines of the National Institute of Health and the 2010/63/EU Directive of the European Parliament on the protection of animals used for scientific purposes. Animal experiments were conducted with the approval of the Landesamt fûr Gesundheit und Soziales (LAGeSo, Berlin) under the internal sacrification license X9010/18, under the institute (MDC) licence X9014/11. The animals had free access to food and water and were kept in individually ventilated cages under a 12 h:12 h light/dark regime (light from 6:30 am to 6:30 pm), a constant $22 \pm 2\,°C$ temperature and $55 \pm 10\%$ humidity.

The process employed enzymatic collagen digestion and retrograde perfusion through the aorta using a Langendorff perfusion apparatus. In brief, the hearts were swiftly removed from the mice after cervical dislocation and connected to a custom-built perfusion system. Initially, the hearts were perfused with perfusion buffer for 4 minutes at a rate of 3 ml/min, followed by an 8-minute perfusion with myocyte digestion buffer containing 5 mg of liberase dispase high DH enzyme (Roche) per digestion. Subsequently, the heart was detached from the perfusion system, and the ventricles were dissected into small pieces using scalpels. After sedimentation, removal of the supernatant, and resuspension the cells were filtered through a nylon mesh cell strainer (100 µm pore size, Falcon) to eliminate any remaining tissue fragments. The cells were then gradually exposed to physiological $Ca^{2+}$ concentrations (~1 mM), resuspended in myocyte plating medium, and seeded on freshly coated Matrigel-coated 8-well Ibidi µ-slides for each experiment. More detailed information about the buffers and materials used can be found in the methods section of Bathe-Peters et al[9].

## hiPSC-CM handling and differentiation

hiPSC-derived cardiomyocytes (hiPSC-CM) were generated according to the protocol previously described[55], and were differentiated for approximately 60 days at the time the experiments were conducted. hiPSCs were purchased from Coriell Institute (GM25256). Briefly, cells were cultured at 37 °C with 5% CO2. The monolayers of hiPSCs were differentiated into hiPSC-CMs by modulating Wnt signaling according to a small molecule-based cardiac differentiation strategy followed by metabolic lactate selection. Differentiated cells were reseeded in 8-well Ibidi µ-slides coated with Geltrex. Imaging was conducted under transmitted light illumination in an inverted Leica DMIRE2 microscope, within a OKOLab sample incubator chamber providing 37 °C, 5% CO2, 85% humidity. Images were acquired using an Andor iXon EMCCD camera operated at 100 Hz, yielding an effective frame time of approximately 26 ms. Kymographs to determine the beating rate were generated using imageJ.

## Cell area measurements with confocal microscopy

To measure the effect of hypotonic treatment on cell area murine cardiomyocytes were seeded in 8-well Ibidi µ-slides with a density of 1,000 cells per well and labeled with Cell Mask™ Deep Red according to manufacturer's protocol. XY and XZ movies of cells were then acquired on a confocal laser scanning microscope, Leica SP8, with a white-light laser at the wavelength of 633 nm and laser power of 5%. All measurements were conducted with an HC PLAP CS2 40×1.3 numerical aperture (NA) oil immersion objective (Leica). Movies were acquired at 30 seconds per frame with a hybrid detector in the range of 643 to 693 nm. Medium change to induce swelling was performed as described in previous sections. ImageJ was used to conduct thresholding and object detection on the cells and the extracted area was plotted with GraphPad Prism v. 9.5.1.

## Linescan measurements with confocal microscopy

HEK293AD cells were seeded in 8-well Ibidi® µ-slides with a density of 25'000 cells per well and transfected with 0.25 µg Gαs-eYFP using the JetPrime® transfection kit according to manufacturer's protocol. Linescans were conducted in a SP8 confocal microscope (Leica

Microsystems) and analyzed as previously described using IgorPro 9 (wavemetrics)[56].

## FRET microscopy

HEK293 cells stably expressing Epac-S$^{H187}$ sensor were seeded in 8-well Ibidi µ-slides with a density of 25'000 cells per well. Cells were washed twice in MES300 and imaged at room temperature. An inverted microscope (DMi8, Leica Microsystems), equipped with an x63 HC PL APO, 1.40-0.60 numerical aperture (NA) objective (Leica Microsystems), dichroic beamsplitter T505lpxr (Visitron Systems), xenon lamp coupled with a continuously tunable Visichrome high-speed polychromator (Visitron Systems) and a metal-oxide-semiconductor camera (Prime95B, Teledyne Photometrics) with a dual image splitter (OptoSplit II, Cairn Research), was used. Excitation wavelength of 445 nm was used, and fluorescence emission was simultaneously recorded at 470/24 nm and 535/30 nm. The movies were obtained at 5 seconds per frame for the number of frames needed for the cAMP concentrations to equilibrate. ImageJ was used to extract fluorescence intensity values from single cells, which were corrected for background and used to calculate FRET/CFP ratio.

## Plate reader cAMP FRET measurements

HEK293 cells stably expressing Epac-S$^{H187}$ sensor were seeded in black 96-well plates (Brand) with a density of 50'000 cells per well. Gα$_s$-KO cells were transfected with either 2.5 µg Epac-S$^{H187}$ or with 1.25 µg Epac-S$^{H187}$ plus 1.25 µg Gα$_s$ 24 h prior to seeding in 96-well plates. 24 h after seeding cells were washed twice with 90 µl MES300 per well. After 5 min incubation at 37 °C, baseline measurement was conducted in a Neo2 plate reader (Biotek) using 420 nm excitation and 485 nm / 540 nm emission filters. As the second step, 60 µl MES300 were removed from each well and replaced with 60 µl of either MES300 or ES150 for the "control" and "swelling" conditions respectively and the plate was measured for 15 minutes. Afterwards the desired dilution series with increasing concentrations of ligand was added (10 µl per well of a 10x concentration in MES300) and the plate was measured again for 15 minutes. At last, a mix of forskolin/IBMX in MES300 was added to an end concentration of 10 µM forskolin and 100 µM IBMX and measured for 10 more minutes. The change in acceptor/donor ratio was normalized to 0% baseline and 100% forskolin/IBMX and plotted in GraphPad Prism v.9.5.1. For the concentration response curves "Dose-response stimulation fit (three parameters)" was applied.

## TIRF microscopy

HEK293AD cells were seeded in 8-well Ibidi® µ-slides with a density of 25'000 cells per well and transfected with JetPrime® transfection according to manufacturer's protocol. Following cDNA amounts per well were used: for Nb80 recruitment – 0.225 µg Snap-β2AR and 0.025 µg Nb80-eYFP; for Nb37 recruitment – 0.05 µg Snap-β2AR, 0.15 µg Gβ$_1$−2A-Gγ$_2$-IRES-Gα$_s$ and 0.05 µg Nb37-eYFP; Transfected cells were labeled with 1 µM SNAP-Surface 647 dye (NEB) for 30 min followed by washing two times for 10 min with 300 µl MES300 per well. After labeling, cells were subsequently taken for imaging to an Attofluor cell chamber (Fisher Scientific) in MES300. A TIRF illuminated Eclipse Ti2 microscope (Nikon), equipped with a x100, 1.49 NA automated correction collar objective and 405-, 488-, 561-, 647-nm laser diodes coupled via an automated N-Storm module and four iXon Ultra 897 EMCCD cameras (Andor), was used. Objective and cell chamber were kept at 37 °C during imaging. The automated objective collar was on, and hardware autofocus was activated. Movies were acquired at 4 s per frame for 400 frames. After a baseline measurement of 50 frames increasing concentrations of isoproterenol were added to the imaged well in 50 frame steps. Prior to each isoproterenol addition 30 µl of solution was removed from the well and 30 µl of 10x isoproterenol in either MES300 or MES200 was applied. ImageJ was used to extract fluorescence intensity values, which were corrected for background and normalized to 0% baseline and 100%

10 μM isoproterenol stimulation. In case of F-actin content measurement HEK293AD cells were transfected with Lifeact-eGFP and imaged as described above at 30 seconds per frame.

## Single-molecule TIRF microscopy

HEK293AD cells were seeded on glass coverslips in 6-well plates at a density of 250,000 cells per well. To ensure low levels of receptor expression, necessary for single-particle imaging, transfection with 2 μg Snap-β2AR was performed with the JetPrime® transfection kit 3-4 hours before imaging. SNAP-tagged receptors were labeled with 1 μM SNAP-surface 549® dye and washed twice for 15 minutes with the imaging buffer (MES300). Imaging was done using AttoFluor chambers with the objective and the sample kept at 20 °C. The movies of receptor diffusion were then recorded via the Cy3 emission channel under illumination by the 561 nm laser at 100% laser power. Exposure time was set to 30 ms and movie length limited to 1000 frames to avoid excessive photobleaching.

## Single particle tracking analysis

Obtained TIRF movies were analyzed by uTrack software[57], loaded to the MATLAB environment. Point source detection and tracking analysis (incl. segment merging/splitting; maximum gap to close – 1 frame) modules were executed and resulting files, containing movie trajectories were loaded to MATLAB. Trajectories were filtered by length (between 3 and 100 frames) and by a manually defined ROI, selecting basolateral membranes of the cells. Mean square displacements for each movie were calculated over a time lag of 100 frames. A linear regression was fit to the first five frames of every resulting MSD movie and the slope was divided by four to obtain the diffusion coefficient.

## Plate reader ligand binding assays

HEK293 cells were seeded in 10 m culture plates, co-transfected with 0.5 μg Snap-β2AR, 0.5 μg Nb37-eYFP and 1.5 μg Gβ$_1$–2A-Gγ$_2$-IRES-Gα$_s$-CFP after 24 h and reseeded in black 96-well plates with a density of 50,000 cells per well another 24 h later. The plate was incubated with 100ul MES300 containing 2 nM JE1319. As the second step, 60 μl MES300 was removed from each well and replaced with 60 μl of either MES300 or ES150 (both containing 2 nM JE1319) for the "control" and "swelling" conditions respectively and the plate was incubated for further 15 minutes. Afterwards the desired dilution series with increasing concentrations of isoproterenol was added (10 μl per well of a 10x concentration in MES300) and the plate was incubated for 90 minutes and measured in a Neo2 plate reader (Biotek) using DualFP polarization filter and an 620 nm/680 nm excitation/emission filter. A G-factor of 0.12 was measured for the instrument. Fluorescence anisotropy was calculated as $r = (I_\parallel - I_\perp)/(I_\parallel + 2I_\perp)$ and plotted in GraphPad Prism v.9.5.1. For the concentration response curves "Dose-response inhibition fit (three parameters) was applied with min and max being constrained to 0% and 100% respectively.

## Plate reader G$_i$-FRET sensor assay

HEK293 cells were seeded in 10 cm cell culture dishes at a density of $4 \times 10^6$. Cells were co-transfected with 0.5 μg SNAP-μOR and 2 μg G$_{i2}$ FRET sensor 24 h after seeding, as described in previous sections; 24 h after transfection, cells were trypsinized and transferred into black 96-well plates (Brand), at a density of 50,000 cells per well; 16 h to 24 h later, cells were washed with MES300, and then 90 μl of MES300 was added to each well. After 10 min incubation at 37 °C, measurement was performed at 37 °C using a Synergy Neo2 Plate Reader (Biotek) using a CFP/YFP FRET filter set. After basal FRET measurement, medium was changed as described above and after 15 minutes of further measurement 10 μl of ligand solution were applied to each well. The change in acceptor/donor ratio was normalized to 0% baseline and plotted in GraphPad Prism v.9.5.1. For the concentration response curves "Dose-response stimulation fit (three parameters)" was applied.

## AlphaScreen cAMP detection assay

HEK293 cells were seeded in transparent 96-well plates (TPP) at a density of 25,000 cells per well. After 24 h the growing medium was exchanged to 100 μ of either MES300 or MES200 and increasing concentrations of isoproterenol were added to corresponding wells. After 10 minutes of stimulation, the plate was frozen at −80 °C for 15 minutes, resulting in cell lysis. The contents of the plate were then thawed at room temperature and 10 μl from each well were respectively transferred to a 384-well white OptiPlate™ plate (Revvity). cAMP standard solutions, acceptor- and donor-bead solutions were prepared and added to the plate according to the manufacturer's protocol. The readouts of fluorescence intensity were performed at a VICTOR Nivo™ plate reader (Revvity) according to a measurement protocol provided by the manufacturer. The resulting cAMP concentrations were calculated and plotted in GraphPad Prism v.10.2.1. For the concentration response curves "Dose-response stimulation fit (three parameters)" was applied.

## Statistical analysis

Statistical analysis was performed in GraphPad Prism v.10.2.1, statistical significance was defined as $P < 0.05$ and the significance values are indicated in the following $P$ value style: 0.1234 (ns), 0.0332 (*), 0.0021(**), 0.0002 (***), <0.0001 (****).

## Reporting summary

Further information on research design is available in the Nature Portfolio Reporting Summary linked to this article.

## Data availability

The source data for the images displayed in Figs. 1–5 and the values used to generate the charts and graphs of Figs. 1–5 are provided as Source Data files. The raw microscopy data underpinning the charts and graphs reported in this manuscript are available under restricted access due to their very large size, and can be obtained by request to the corresponding author. Please allow for two weeks to address the request, and data will be shared via an appropriate online repository for the duration of a week. Source data are provided with this paper.

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

## Acknowledgements

This project was supported by the Deutsche Forschungsgemeinschaft (DFG), via collaborative research center 1423 Project 421152132, Sub-project C03 (to P.A. and M.J.L.). P.A. would like to acknowledge generous funding by the Leverhulme Trust via RL-2022-015. A.I. was funded by the Japan Society for the Promotion of Science (JP21H04791 and JP24K21281), the Japan Agency for Medical Research and Development (JP22ama121038 and JP22zf0127007), and the Japan Science and Technology Agency (JPMJFR215T and JPMJMS2023). J.L.M.K. acknowledges a World Leading PhD fellowship from the Universtiy of St Andrews. We are grateful to Bärbel Pohl, Marlies Grieben and Ulrike Zabel for their excellent technical assistance. We would like to acknowledge Heike Biebermann (Charité UniversitätsMedizin, Berlin) for kindly sharing the HEK293 cells stably expressing Epac1-S$^{H187}$, helping with alphaScreen assay and Peter Gmeiner (Friedrich-Alexander-Universität, Erlangen) for synthetizing the ligand JE1319. Nb80 and Nb37 plasmids were a gift from Mark von Zastrow (UCSF). Epac1-S$^{H187}$ biosensor plasmid was a gift from Kees Jalink (Addgene plasmid #170348; http://n2t.net/addgene:170348; RRID:Addgene_170348). We are grateful to Romy Thomas (Max Delbrück Center, Berlin) for helping us generate the tricistronic heterotrimeric G protein expression vector used in our experiments.

## Author contributions

A.S. and P.A. designed research; A.S. performed research; A.I. contributed new reagents/analytic tools; A.S. and P.A. analyzed data; M.B.-P. supported the preparation of primary cardiomyocytes; J.L.M.K. prepared and imaged hiPSC-derived CMs. P.A. and M.J.L. supervised the project and acquired funding; P.A. and A.S. wrote the paper with input from all authors.

## Competing interests

The authors declare no competing interests.
