## [Peer Review File · Nature Communications]

Cell swelling enhances ligand-driven β -adrenergic signalingReviewers' comments:

Reviewer #1 (Remarks to the Author):

While this is an interesting observation, similar phenomena have been reported previously, in which deletion/disruption of lipid rafts and/or caveolin can enhance GPCR (i.e., beta2AR) signaling in mammalian cells. The overall conclusion is premature. Several major concerns are listed here.

1. One of the fundamental issues with the study is the data processing and presentation. All the cAMP FRET data are normalized against the FSK/IBMX. This is based on an underlying assumption that the FSK/IBMX induced similar increases in cAMP FRET ratio in both swelling and control cells, which may not be true. For example, in pathological conditions, the FSK/IBMX-induced responses are often reduced in failing cardiac myocytes than in control cells. The raw traces normalized against the baseline should be presented.
2. Similarly, after normalizing against the saturated dose of 10uM ISO, the data in Figures 4 and 5 show more recruitment of downstream signaling proteins at lower doses of isoproterenol. The authors concluded that the cell swelling increased GPCR coupling to downstream G protein and arrestin. This conclusion may be affected if the maximal increases were different. Additionally, the underlying mechanisms remain unclear. For example, the cell surface receptor density may be increased in swelling cells, which may lead to higher responses.
3. Caveolae represent the most significant curvature in mammalian cells, whereas the lipid rafts represent the rigid membrane structure that is enriched with receptors and signaling molecules. What if the authors treat cells with drugs to deplete lipid rafts and caveolae? Do they still observe similar impacts with swelling?
4. In Figure 3C, the swelling cells appear to have an increased cAMP FRET ratio at the baseline, which could abrogate the small difference in Figure 3f. If true, the data would suggest that cAMP degradation is also affected by swelling. The increased baseline could also significantly affect the small differences in the EC50 of isoproterenol in inducing cAMP in Figure 1.
5. In Supp Figure 3d, FSK induced higher cAMP in swelling cells than controls, suggesting that AC activation is affected by swelling independent of changes in receptor activity. It is very confusing why this difference was lost in Gs-KO cells but restored in Gs-expressing cells. Again, the raw values without normalizing against FSK/IBMX should be plotted. The author stated, "The diterpene forskolin and G α s have a synergistic/ cooperative action in activating ACs." but did not provide a reference. Do authors argue that the action of FSK is dependent on G proteins?
6. The arrestin, NB37, and NB80 data have similar issues. Does 10uM ISO induce similar maximal increases in the recruitment of these signaling proteins? If not, then the absolute increases induced by low doses of ISO may be similar? which may lead to a completely different conclusion.
7. The difference in Ki is marginal in Figure 5D and Supp. Figure 4. At a -9.3 dose in Figure 5D, there is a big difference between swelling cells and control cells. If normalized against the values at this dose, do the authors observe the small difference in Ki? Beta2AR has shown two binding affinities: a higher binding affinity at the nM range and a lower affinity at the uM range. Based on the overall observations of the effects at nM doses of ISO, the high-

affinity binding site should be affected. Do authors observe anything in the context?
8. Additionally, it is not clear whether this subtle difference in K_i is sufficient to explain the signaling difference in NB80, NB37, and arrestin recruitment, for example, in Figure 5e.
9. The data in Supp Figure 6 is more than confusing. Does swelling decrease the receptor affinity to antagonist Prop? If so, please provide evidence. Even so, it is not clear why 100 μ M Prop cannot inhibit 1 μ M ISO response in swelling cells.

Reviewer #2 (Remarks to the Author):

This is an interesting report which claims to have discovered a new mode of GPCR regulation by osmotic cell swelling. This idea suggests that changes to cell environment can impact on receptor function in terms of ligand binding and activation of downstream signaling, as exemplified by beta2 adrenergic receptor (bAR) - cAMP signaling pathway. Although the idea is indeed interesting, I could detect one major and one minor limitation which should be addressed by the authors to achieve the level of priority needed for publication in a high impact journal.

Major. The whole message of the study relies almost completely on the new imaging techniques based on fluorescent biosensors/probes to detect receptor activation, G-protein/arrestin recruitment, cAMP synthesis and even measuring active receptor conformation using fluorescently tagged nanobody. These techniques are indeed powerful, allow real time monitoring of receptor activation cascade and are widely used these days to study GPCR signalling. However, those biosensors are prone to well known artefacts- cell movement (relevant of swelling), pH sensitivity, etc. The authors have offered just one "control" experiments stimulating cell with 8-Br-cAMP-AM (sFig 1B) and showing that dose-response curve was not much affected. This control experiment has several limitations - (i) this is a biosensor based experiment, (ii) it bypasses receptor activation looking purely at the sensor activation directly by the cAMP analogue. At the same time, many FRET traces in response to ISO show an initial abrupt increase of FRET ratio upon ISO infusion which is strange in this context (here I would really recommend to show concomitant individual CFP and YFP traces in parallel). Therefore, to justify the notion that receptors, G-proteins, cAMP etc are directly affected by swelling, it is indispensable or better said absolutely mandatory to perform a range of classical biochemical assays- at least radioligand receptor binding assays and cAMP accumulation assays (RIA or ELISA) to demonstrate that swelling indeed significantly changes receptor affinity for the ligand and cAMP generation, matching real time assay results.

Minor. The authors tend to extrapolate a lot (to much to be completely honest) that this mechanism may apply to other GPCRs and cells types without unequivocally demonstrating this statement. For example, their experiment in Fig 2 shows cAMP measurements in mouse cardiomyocytes isolated from CAG-Epac1 transgenic mice (ref 9 supposed to cite these cells/mice has no information about this transgenic line). It is well known from the literature that beta1 adrenoceptors contribute to roughly 80-90% of cAMP response to ISO and only a minor fraction is due to beta2 receptors. Are we monitoring the beta2 or beta1 response here? How about disentangle individual beta1 and beta2 contributions by combining ISO

with beta1 vs beta2 inhibitor (CGP20712A and ICI118551, respectively)? Moreover, I was somehow missing controls in this and HEK cell experiments with beta2 antagonist/inverse agonist to show that it blocks the effect of swelling on GPCR cascade activation. In terms of general concepts and translational relevance, how does the pathology (e.g. ischemic injury mentioned by authors) would affect their measurements directly? Would other ways of affecting cell/myocytes size such as stretching would have similar effects?

Reviewer #3 (Remarks to the Author):

The study by Sirbu and colleagues investigates the effect of hypoosmotic stress on the signaling of β 2-adrenergic receptors (β 2-AR) in living cells. The authors identify that cell swelling leads to an increase in the isoproterenol-driven cAMP response in HEK293 cells (Fig. 1) and cardiomyocytes (Fig. 2). Increased cAMP levels are also observed after activation of the Gs-coupled histamine and melanocortin receptors, and the response of the Gi-coupled opioid receptors is also enhanced by cell swelling. The authors subsequently perform several experiments aiming to provide mechanistic understanding of the enhanced cAMP transients upon β 2-AR activation and identify a role for Gs in the response to cell swelling (Fig. 3). Using protein relocalization assays, the authors then show an increased recruitment of the Gs activation sensor Nb37 and a faster recruitment of β -arrestin2 to the plasma membrane in hypoosmotic stress conditions (Fig. 4). Finally, the authors present evidence for an increased isoproterenol affinity for β 2-AR and a left-shifted concentration response curve for β 2-AR activation (Nb80 recruitment) in hypotonic conditions (Fig. 5). Based on the results, the authors propose the model that cell swelling favors the active β 2-AR conformation, which would enhance G protein coupling and thereby an increase in cAMP production.

The study addresses an important research question, namely how mechanical forces effect intracellular biochemical signals, which is an emerging topic in GPCR biology and could be relevant to various physiopathological conditions. However, the experimental data are limited, often indirect, and do not offer a molecular explanation for the effect of cell swelling on GPCR activation and cAMP signaling. Therefore, the hypothesis that osmotic swelling directly impacts β 2-AR's conformation and ternary complex formation appears speculative based on the data presented in this study. Several key open questions remain and alternative explanations for the observed cAMP response increase need to be ruled out. Furthermore, it is not clear how the cell swelling conditions relate to physiological settings. In addition, the authors do not acknowledge existing studies on class A GPCRs, which have already investigated the effect of osmotic shock on receptor signaling (see below).

The major concerns are listed below:

1. Cell swelling/hypotonic solution induces a reduction of clathrin-mediated endocytosis (e.g. PMID: 28923837 & 35389747). The authors should probe how cell swelling affects the trafficking and desensitization process of β 2-AR. It is possible that elevated receptor levels and/or a block in endocytosis are underlying the changed cAMP profile. The authors should carefully measure the effect of cell swelling on cell surface β 2-AR levels, i) before agonist (to determine possible alterations in baseline receptor levels) and ii) upon agonist treatment.
2. Cell swelling at 200 mOsm is performed in all presented experiments, however the

rationale for the chosen osmolarity and time point are not mentioned. It appears that cell swelling continues linearly with time (Fig. 2b). Are the experimental conditions relevant to (patho-)physiological conditions? How robust is the effect on the cAMP response across hypotonic stress conditions, i.e. how is signaling affected at various osmolarities and at different time points upon switch to swelling media?

3. Cell swelling results in increased plasma membrane tension (e.g. PMID: 34785592), yet the authors do not provide any experiment / do not discuss the possibility that surface tension affects the activation of β 2-AR/Gs. It is already well established that multiple GPCRs can be activated by mechanical stimuli including cell stretching, shear stress, and hypotonic shock (mentioned here as cell swelling) (e.g. PMID: 31857598, 17030791, 25170081, <https://www.sciencedirect.com/science/article/pii/S2468867323000597#bib48>). The known role of membrane tension on GPCRs / β 2-AR must be mentioned and discussed in introduction and discussion. Furthermore, if the authors think that membrane tension does not contribute to the here observed potentiation of the Iso effect, additional experiments need to prove that membrane tension is not affected upon cell swelling (e.g. using Flipper-TR membrane tension probes) and that parallel ways to increase membrane tension (e.g. cell stretching) do not promote a similarly enhanced cAMP response.

4. The authors show enhanced AC activity in cell swelling conditions independently of β 2-AR activation, which is dependent on Gs (Fig. 3d). How do the authors explain this result? Does cell swelling drive an elevated basal Gs tone? What is the mechanism and how does it influence the β 2-AR pathway studied in the other experiments?

5. The observed increase in transducer protein recruitment upon cell swelling is based on TIRF experiments, which measure relocalization of proteins from the cytosol to the plasma membrane in close contact with the glass cover slip. Given that cell swelling leads to a flattening of the membrane (described by the authors as 'ironing out wrinkles in a cloth' in the introduction), the TIRF signal may directly be affected due to changes in membrane topology and may underlie the detected transducer protein signal increase. The authors need to control for such a possibility by including intensity traces of PM localized or PM-recruited control proteins that do not show an altered signal.

6. The observed change in arrestin recruitment upon cell swelling does not seem to be an increase in recruitment but rather a faster recruitment (Fig. 4g). How do the authors explain that G protein activation is increased (Fig. 4c) and arrestin recruitment accelerated? Is cell swelling affecting the diffusion of transducer proteins differently? The authors mention that protein diffusion is not modified in the cell swelling condition, yet the data is not included in the manuscript. The data should be added.

7. It is a tempting hypothesis that increased membrane tension favors an intermediate state closer to the active GPCR conformation, with increased affinity of β 2-AR for Iso and decreased affinity for an antagonist, which is briefly mentioned in the discussion, with data presented in Supp. Fig 6. Why do the authors 'hide' the data in the supplementary material?

Reviewer #4 (Remarks to the Author):

Sirbu et al. demonstrate that cellular osmotic swelling potentiates β 2AR ligand-mediated acute cytosolic and plasma membrane cAMP generation in HEK293 cells, as well as cytosolic cAMP in adult mouse ventricular cardiomyocytes. This swelling-mediated effect was also observed upon stimulation with ligands selective for distinct Gs and Gi-coupled endogenous

receptors in HEK 293 cells. Additionally, using the HEK 293 cell model, the authors demonstrate that the increased ligand potency for cytosolic cAMP generation with osmotic swelling persists with inhibition of either receptor internalization or phosphodiesterase activity, and with direct stimulation of adenylyl cyclase. However, Gs protein knockdown prevented the osmotic swelling-induced cAMP potentiation with adenylyl cyclase activation. This was consistent with increased Nb37-mYFP recruitment to membrane upon β AR stimulation and osmotic swelling, suggesting that swelling increases Gs-receptor association. Similar results are observed for β -arrestin recruitment. Through fluorescent ligand displacement experiments and Nb80 recruitment assays, the authors show that swelling increases receptor-ligand affinity and promotes an active receptor conformation. While the concept of osmotic swelling-induced changes in receptor allostery is not new (for instance, see PMID 25170081), this is the first such study to quantify osmotic swelling-induced changes in ligand potency on cAMP generation in response to β 2AR stimulation.

Comments

1. While this is a well-designed molecular pharmacology study characterizing the phenotype of the potentiation of cAMP signaling, it does not offer biophysical/mechanistic insight into what structural changes are induced by osmotic stress or domains/amino acid residues of β 2AR required for the effect to which water molecules presumably bind.
2. Most of the study focuses on the osmotic swelling-induced increase in ligand potency of the Gs-cAMP effect, but Fig4f-h indicates enhanced β -arrestin recruitment as well. Since ternary complexes containing β 2AR, G protein (s or i) and β -arrestins have been shown, is this effect also dependent on G protein expression?
3. The authors show osmotic stress also increases cAMP generation in response to the same level of ligand stimulation in adult cardiomyocytes. There is substantial literature describing compartmentation of β AR signaling in cardiomyocytes that relays distinct functional outcomes, does osmotic stress alter this compartmentation? The authors suggest these effects could relate to how receptors in cardiomyocytes may respond to ischemic injury but have not followed through to determine if there is a functional effect on the cardiomyocytes either by osmotic stress alone or in combination with ischemic stress.
4. Dyngo-4 treatment on its own reduced both the potency and efficacy of ISO-mediated cAMP generation; does this suggest that the cytosolic cAMP generation being measured is primarily due to endosomal receptor activation that is altered by Dyngo-disrupted receptor trafficking, and does the osmotic swelling correct for that independent of a direct structural change in the receptor?

Reviewers' comments:

Reviewer #1 (Remarks to the Author):

While this is an interesting observation, similar phenomena have been reported previously, in which deletion/disruption of lipid rafts and/or caveolin can enhance GPCR (i.e., beta2AR) signalling in mammalian cells. The overall conclusion is premature. Several major concerns are listed here.

1. One of the fundamental issues with the study is the data processing and presentation. All the cAMP FRET data are normalized against the FSK/IBMX. This is based on an underlying assumption that the FSK/IBMX induced similar increases in cAMP FRET ratio in both swelling and control cells, which may not be true. For example, in pathological conditions, the FSK/IBMX-induced responses are often reduced in failing cardiac myocytes than in control cells. The raw traces normalized against the baseline should be presented.

We thank the reviewer for this remark. We had adopted this layout since we had observed that the non-normalized FRET response to FSK/IBMX was the same in control or swollen cells, as clearly displayed below in **Figure R1 a**. **Figure R1 b,c** show side to side the normalized and non-normalized responses in terms of FRET ratio. The effect of swelling on the cAMP peak is clearly the same, independent of the normalization. The respective donor and acceptor traces show the expected antiparallel behaviour, and again the effect of swelling clearly persists (**Figure R1 d**).

Fig. R1 Cell swelling increases cAMP production in response to isoproterenol stimulation, while not affecting cAMP response to saturating stimulus of forskolin/IBMX
a average kinetic curve of acceptor/donor ratio measured in HEK293T cells stably expressing Epac-S^{H187} in a microplate reader at baseline, after medium change to control of swelling medium and after addition of 10 μM Forskolin + 100 μM IBMX; **b-d** representative kinetic curve of acceptor/donor ratio measured in HEK293T cells stably expressing Epac-S^{H187} in a

microplate reader (**b** - normalized to baseline and 10 μ M Forskolin + 100 μ M IBMX, **c** – non-normalized ratio, **d** – respective CFP and YFP channel traces).

2. Similarly, after normalizing against the saturated dose of 10 μ M ISO, the data in Figures 4 and 5 show more recruitment of downstream signalling proteins at lower doses of isoproterenol. The authors concluded that the cell swelling increased GPCR coupling to downstream G protein and arrestin. *This conclusion may be affected if the maximal increases were different.* Additionally, the underlying mechanisms remain unclear. For example, the cell surface receptor density may be increased in swelling cells, which may lead to higher responses.

We are grateful for this comment from the reviewer. The first part of the comment is addressed under **point 6**. In regard to the cell surface receptor density, we have used molecular brightness, an accurate method to quantify receptor density at the plasma membrane (Isbilir A... Annibale P. Nature Protocols 2021) based on confocal micrograph of the basolateral membrane of receptor-expressing cells (**Figure R2a**) to verify surface receptor density in swollen vs non-swollen cells both before (**Figure R2b**) as well as after (Figure R2c) isoproterenol addition for 10 minutes. Receptor density and receptor membrane expression after internalization (across dozens of cells tested) are clearly not affected, and thus are not the origin of the observed effects due to swelling.

Fig. R2 Swelling does not affect basal number of β 2AR available at the plasma membrane and does not affect isoproterenol-mediated internalization

a representative confocal images of basolateral membrane of HEK293AD cell transiently transfected with Snap- β 2AR and labelled with SnapSurface-Alexa674 dye before and after hypotonic treatment; **b** number of receptors per pixel as obtained from single cells before and after hypotonic treatment (paired two-tailed t-test; mean \pm SEM; n =32 cells from 4 independent experiments); **c** relative number of receptors at the plasma membrane (compared to baseline) after 10 min of 10 μ M isoproterenol stimulation (unpaired two-tailed t-test; mean \pm SEM; n = at least 23 cells from 3 independent experiments).

3. Caveolae represent the most significant curvature in mammalian cells, whereas the lipid rafts represent the rigid membrane structure that is enriched with receptors and signaling molecules. What if the authors treat cells with drugs to deplete lipid rafts and caveolae? Do they still observe similar impacts with swelling?

Despite the emerging importance of membrane microdomains for cellular signaling, we have not addressed 'lipid rafts' explicitly in our investigation due to the significant debate surrounding their features and physiological role (10.1038/nrm.2017.16, 10.1016/j.tcb.2020.01.009). The only report we could find to date connecting lipid organization and cell signalling upon swelling conditions in live cells is a decade-old work on intermediate conductance (IK) channels doi.org/10.1074/jbc.M607730200. Nonetheless, we agree with the reviewer that cell swelling may affect membrane lipid composition. Osmotic tension was reported to induce phase separation in lipid vesicles (10.1021/acs.langmuir.9b03893). Furthermore, upon prolonged and more extreme swelling, increase in membrane surface area needs to be supported by recruitment of additional phospholipids to the membrane (10.1007/s00232-006-0080-8). Another study demonstrates, how prolonged exposure to osmotic stress leads to a reorganization of the plasma membrane structure in plant cells (10.1007/s00232-021-00194-x).

However, crucially, both the timescale of our experiments as well as the magnitude of the swelling we induced are significantly milder than these conditions.

Caveolae are indeed recognised as important mediators of membrane mechanical response to external stresses (10.1016/j.cell.2010.12.031), and their role in adrenergic signalling (10.1016/j.yjmcc.2004.04.018) is also well established, especially in relation to beta2-AR signalling. Rasenick and colleagues reported that caveolae (disrupted by hyposmotic swelling) attenuate Gs signaling in response to isoproterenol (10.1124/mol.109.060160). However, methyl- β -cyclodextrin (M β CD) treatment did not alter basal cAMP levels.

Osmotic swelling was reported to specifically disrupt Caveolin1 – Gq protein interaction and attenuate Gq-mediated Ca²⁺ release (10.1074/jbc.m115.655126). In this work, the same effect on Gq signalling is observed upon cholesterol depletion under M β CD treatment, which leads to perturbation in caveolae structure. Interestingly, the same treatment was reported to increase (10.1039/B823243A) beta-adrenergic mediated response to isoproterenol. This would be in line with the observation that cell swelling leads to a reduction in number of caveolae (10.1016/j.cell.2010.12.031). Overall, there is thus a large and often conflicting mole of data on the effects of long-term/permanent modifications of membrane microdomains on GPCR signalling, but none addressed the effects on cAMP caused by mild hyposmotic swelling, which in our observations are transient. Moreover, our results show that a similar swelling-induced response is observed, besides the beta-AR system, also for receptors that are not traditionally associated to caveolae, such as histamine receptors or melanocortin receptors (**Supplementary Figure 1 F-H**).

To address the question from the reviewer, we investigated the effect of pre-treating cells with M β CD. We still observe an effect of osmotic swelling on isoproterenol mediated cAMP increase, indicating the broad independence of our effect from membrane composition or caveolar integrity (**Figure R3**).

Fig. R3 Swelling increases isoproterenol mediated cAMP production under cholesterol depletion

a representative curve of intracellular cAMP concentration with and without pretreatment with 10 mM methyl-β-cyclodextrin for 60 minutes; **b** averaged concentration response curves representing maximal cAMP concentrations (mean ± SEM; n = 3 independent experiments).

4. In Figure 3C, the swelling cells appear to have an increased cAMP FRET ratio at the baseline, which could abrogate the small difference in Figure 3f. If true, the data would suggest that cAMP degradation is also affected by swelling. The increased baseline could also significantly affect the small differences in the EC₅₀ of isoproterenol in inducing cAMP in Figure 1.

We thank for the reviewer for pointing this out. Indeed, in the representative trace of **Figure 3C** there is a slight increase in baseline upon change to the swelling medium. Slight changes are routinely observed upon medium change to the plate. However, across all our experiments, we have not observed any average change to the baseline upon swelling, as can be clearly seen from the concentration response curve displayed in **Figure 1D** and additionally from the **Figure R4**. Our effects are sizable and consistent at the level of EC₅₀, as clearly shown in **Figure 3F**. This observation sets our work apart from any claim of a curvature-driven constitutive receptor activation, since the effects that we report are exclusively seen in the context of a potentiation of agonist response.

Fig. R4 Cell swelling doesn't affect basal cAMP levels in HEK293 cells.

a average cAMP changes after medium replacement **b** pairwise comparison of FRET ratios between control and swelling condition during baseline and after medium replacement (paired t-test; for **a** and **b** – mean ± SEM; n = 11 independent experiments); **c** representative curve of

intracellular cAMP concentration under inhibition of PDE activity by 100 μ M IBMX without isoproterenol stimulation.

5. In Supp Figure 3d, FSK induced higher cAMP in swelling cells than controls, suggesting that AC activation is affected by swelling independent of changes in receptor activity. It is very confusing why this difference was lost in Gs-KO cells but restored in Gs-expressing cells. Again, the raw values without normalizing against FSK/IBMX should be plotted. The author stated, "The diterpene forskolin and Gs have a synergistic/ cooperative action in activating Acs." But did not provide a reference. Do authors argue that the action of FSK is dependent on G proteins?

Indeed, Forskolin response is enhanced in swollen cell. However, the Gs-KO knockout experiments allowed us to rule out, early on, a specific origin of the observed effects at the AC. The Gs has a synergistic effect with FSK at AC2, AC4, AC5, AC6, and interaction with Gs is seen as a prerequisite for a high-affinity binding site for FSK. (10.1126/science.278.5345, 10.1074/jbc.272.35.22265, 10.1016/j.neuropharm.2005.11.004 and 10.1146/annurev.pa.36.040196.002333, 10.1023/A:1023684503883 and references therein). We apologise for having omitted, in one of the several revision steps, these references regarding the synergistic effect of Forskolin and Gs.

As discussed in reference to point 1, the enhanced response to FSK does not affect in any way the conclusions to be drawn from our observations. Moreover, even if a larger FSK response were used to normalize our swelling traces, this should rather blunt, rather than amplify, the normalized amplitude of the cAMP transients we see.

6. The arrestin, NB37, and NB80 data have similar issues. Does 10uM ISO induce similar maximal increases in the recruitment of these signaling proteins? If not, then the absolute increases induced by low doses of ISO may be similar? Which may lead to a completely different conclusion.

We thank the reviewer for this comment. 10 μ M ISO indeed induces maximal recruitment for Nb37 and β -arrestin2, as shown in **Figure R5** below. For the Nb80 a saturation in recruitment is reached at the isoproterenol concentration of 100nM, as seen in **Figure 5e** of the original manuscript. Given single cell variability we chose to normalize each cell recruitment experiment to its own saturating value (10 μ M ISO). The effect of swelling is then appraised not by comparing maximal stimulation, but again concentration response curves (for Nb80) or non-saturated response, that highlight unambiguously the enhancement provided by swelling on the recruitment of both partners.

Fig. R5 10 μ M isoproterenol saturate Nb37-eYFP and β -Arr2-eYFP recruitment

a,b representative curves showing relative increase of membrane fluorescence collected after 1, 10 and 100 μM isoproterenol addition respectively in cells transfected with Snap- β2AR and Nb37-eYFP (**a**) or $\beta\text{-Arr2-eYFP}$ (**b**)

7. The difference in K_i is marginal in Figure 5D and Supp. Figure 4. At a -9.3 dose in Figure 5D, there is a big difference between swelling cells and control cells. If normalized against the values at this dose, do the authors observe the small difference in K_i ?

The difference in K_i values that the reviewer refers to, graphically reported in **Figure 5C** (not D), has a p-value lower than 0.05, that makes it significant. We are unsure why the reviewer refers to this as marginal, in this instance.

We are also unsure why the reviewer would like to normalise both graphs to -9.3 , since in this case the lower anisotropy of the swollen cells at -9.5 (not -9.3) could be just an outlier. In any case, to address the matter fully we show here the non-normalized curves and the respective fits, that still show a significant effect of swelling on the K_i (**Figure R6**).

Fig. R6 Non-normalized fluorescence anisotropy curves show an increase of isoproterenol binding affinity in swollen cells

a non-normalized average fluorescence anisotropy curves of isoproterenol displacing 2 nM of fluorescently labelled JE1319 from the $\beta\text{2-AR}$ in swollen and non-swollen cells, together with unconstrained fit; **b** scatter plot comparing $\log(K_i)$ values obtained from single binding experiments; **c** pairwise comparison of $\log(K_i)$ values represented in **b**, and the average of the mean of differences (mean \pm SEM; $n = 6$ plates from 5 independent experiments; statistical analysis by a paired two-tailed t-test).

Beta2-AR has shown two binding affinities: a higher binding affinity at the nM range and a lower affinity at the μM range. Based on the overall observations of the effects at nM doses of ISO, the high-affinity binding site should be affected. Do authors observe anything in the context?

We thank the reviewer for their comment. Our understanding is that while a bimodal ligand binding has been observed for GPCRs (with and without G protein), this has been typically observed in membranes (eg. by radioligand binding), and it is very hard to detect this in intact cells experiments. For this reason, we did not feel like speculating about the two affinity states, and how swelling may affect them separately.

8. Additionally, it is not clear whether this subtle difference in K_i is sufficient to explain the signaling difference in NB80, NB37, and arrestin recruitment, for example, in Figure 5e.

Again, we politely disagree with the reviewer The shift in K_i for ligand binding is **0.12 of a Log unit**. The shift in EC_{50} for NB80 recruitment is **0.3 of a Log unit**, so they are comparable. We did not calculate EC_{50} s for NB37 or beta-arrestin2 recruitment.

9. The data in Supp Figure 6 is more than confusing. Does swelling decrease the receptor affinity to antagonist Prop? If so, please provide evidence.

Even so, it is not clear why 100 μ M Prop cannot inhibit 1 μ M ISO response in swelling cells.

In our Supplementary Figure 6, we observed that swelling enhanced substantially recruitment of NB80 to the receptor, even in cells pre-treated with propranolol. In this case, 1 μ M ISO, was sufficient to break a 100 μ M PROP blockade. Assuming a dissociation constant K_D for isoproterenol $K_D^{ISO}=150$ nM and a $K_D^{PROP}=2.6$ nM, at relative concentrations of 1 μ M ISO and 100 μ M PROP, assuming they are in large excess of the number of receptor sites available, one would expect full occupancy by propranolol. Thus, our results point either to a ligand-independent conformational changes that allows NB80 recruitment despite propranolol, or to an overall conformational change that substantially reduces K_D of propranolol (e.g. by increasing K_{OFF}), allowing instead binding by ISO. We expanded our investigation to monitor also downstream signaling and applied this experimental configuration (pre-incubation with PROP followed by ISO stimulation with/without swelling) to measure downstream cAMP production. Interestingly, in control cells, only stimulations more than 1 μ M ISO were able to partially break the blockade and led to cAMP production, whereas in swollen cells a transient peak was observed also for 10 nM and 100 nM ISO concentrations (**Figure R7**), suggesting that upon the swelling phase we observe a transient decrease of affinity towards PROP and/or increase in affinity towards ISO, in line with the results presented in the manuscript.

Fig. R7 Cell swelling increases cAMP production in response to isoproterenol stimulation under receptor inhibition by a prototypical β -blocker propranolol

Representative kinetic curves of acceptor/donor ratio measured in HEK293T cells stably expressing Epac-S^{H187} in a microplate reader after 1h preincubation with 100 μ M propranolol and consecutive addition of indicated isoproterenol concentrations; cells were additionally incubated in control (left) or swelling (right) medium, containing 100 μ M propranolol, before isoproterenol addition.

Reviewer #2 (Remarks to the Author):

This is an interesting report which claims to have discovered a new mode of GPCR regulation by osmotic cell swelling. This idea suggests that changes to cell environment can impact on receptor function in terms of ligand binding and activation of downstream signaling, as exemplified by beta2 adrenergic receptor (bAR) - cAMP signaling pathway. Although the idea is indeed interesting, I could detect one major and one minor limitation which should be addressed by the authors to achieve the level of priority needed for publication in a high impact journal.

Major. The whole message of the study relies almost completely on the new imaging techniques based on fluorescent biosensors/probes to detect receptor activation, G-protein/arrestin recruitment, cAMP synthesis and even measuring active receptor conformation using fluorescently tagged nanobody. These techniques are indeed powerful, allow real time monitoring of receptor activation cascade and are widely used these days to study GPCR signalling. However, those biosensors are prone to well know artefacts- cell movement (relevant of swelling), pH sensitivity, etc. The authors have offered just one "control" experiments stimulating cell with 8-Br-cAMP-AM (sFig 1B) and showing that dose-response curve was not much affected. This control experiment has several limitations - (i) this is a biosensor based experiment, (ii) it bypasses receptor activation looking purely at the sensor activation directly by the cAMP analogue. At the same time, many FRET traces in response to ISO show an initial abrupt increase of FRET ratio upon ISO infusion which is strange in this context (here I would really recommend to show concomitant individual CFP and YFP traces in parallel). Therefore, to justify the notion that receptors, G-proteins, cAMP etc are directly affected by swelling, it is indispensable or better said absolutely mandatory to perform a range of classical biochemical assays- at least radioligand receptor binding assays and cAMP accumulation assays (RIA or ELISA) to demonstrate that swelling indeed significantly changes receptor affinity for the ligand and cAMP generation, matching real time assay results.

We are grateful to the reviewer for pointing out these methodological aspects. In our answer to Reviewer #1, **Figure R1b-d** we show clean antiparallel traces from our FRET biosensor measurements. As a matter of fact, this sensor is ratiometric, i.e. relies on the ratio between two separate spectral channels: this already allows to compound several of the concerns of the reviewer (i.e. defocus, or sample movement). For single color experiments (i.e. using Nb recruitment) we have shown Movies (e.g. Supp Movie 2) and before/after TIRF snapshots, to prove that sample did not move, or defocus during imaging. As a matter of fact, the TIRF microscope that we used has an intrinsic autofocus system (Nikon perfect focus system), which maintains the sample in focus also over long acquisitions.

Concerning the classical biochemical assays, we would like to point out that since Roger Tsien introduced the first fluorescence Ca²⁺ sensors based on EGFP-like proteins, fluorescence – based biosensing has been used and validated by a substantial number of world-leading groups. Nonetheless, we agree that the measurements suggested by the reviewer would provide independent controls, in particular global cAMP measurements via a biochemical assay. On the other hand, radioligand binding would need to be conducted in whole cells, to enact swelling.

We have now conducted an endpoint cAMP readout assay, which we display below (**Figure R8**). Untransfected HEK293T cells were placed in isoosmotic or hypoosmotic medium containing 1mM IBMX and stimulated with a range of isoproterenol concentrations for 10 minutes. After this, cells were lysated by freezing them at -80°C for 15 minutes and the lysates were tested using the AlphaScreen cAMP detection kit (with kind support from the group of

Prof Heike Biebermann). The results of these measurements clearly show a much higher accumulation of cAMP in swollen cells, with saturation being reached already at stimulation with 100nM isoproterenol. Furthermore, we do not observe any significant differences in basal cAMP concentrations between the two conditions. These results are perfectly in line with the results obtained with the FRET cAMP biosensor.

Fig. R8 Endpoint cAMP measurement shows a strong shift in isoproterenol potency in swollen cells.

a Concentration-response curve obtained by measuring cAMP concentrations with the AlphaScreen cAMP assay in cell lysates of untransfected HEK293T cells, that were incubated in an isotonic (red) and hypotonic (blue) medium containing indicated concentrations of isoproterenol for 10 minutes prior to cell lysis; **b** $\text{log}(EC_{50})$ values obtained on independent days (mean \pm SEM; n = 2 independent experiments).

Minor. The authors tend to extrapolate a lot (too much to be completely honest) that this mechanism may apply to other GPCRs and cells types without unequivocally demonstrating this statement. For example, their experiment in Fig 2 shows cAMP measurements in mouse cardiomyocytes isolated from CAG-Epac1 transgenic mice (ref 9 supposed to cite these cells/mice has no information about this transgenic line). It is well known from the literature that beta1 adrenoceptors contribute to roughly 80-90% of cAMP response to ISO and only a minor fraction is due to beta2 receptors. Are we monitoring the beta2 or beta1 response here?

Indeed, most cAMP response in HEK293 cells comes from beta1-AR stimulation. Our results from Figures 1-4 inclusive, are in fact representative of the effect of both receptors. We further proved that the effect is also present for the 'minority' beta2-AR by using selective agonists as reported in **Supplementary Figure 1 c-e**. Furthermore, we have conducted experiments in CHO cells, which lack an endogenous β -adrenergic background, by transfecting either beta1- or beta2-AR. The results show that beta1-AR and beta2-AR mediated cAMP signaling is affected by cell swelling in a similar manner (**Fig. R9**).

β1AR overexpression

β2AR overexpression

Fig. R9 Effect of swelling on receptor signaling is applicable to both β1- and β2AR and can be reproduced in CHO cells

a,c representative curves of intracellular cAMP concentration upon stimulation with 10 pM isoproterenol of CHO cells transfected with Snap-β1AR (or Snap-β2AR for **c**) and Epac-S^{H187} sensor **b,d** Averaged concentration response curves of max cAMP concentrations obtained under same conditions as **a** and **c** respectively (mean ± SEM; n = at least 2 plates from 2 independent experiments)

We apologise about an error in identifying the appropriate reference for the transgenic CAG-epac1 line, which is actually 10 (10.1371/journal.pbio.1000172).

How about disentangle individual beta1 and beta2 contributions by combining ISO with beta1 vs beta2 inhibitor (CGP20712A and IC118551, respectively)?

Moreover, I was somehow missing controls in this and HEK cell experiments with beta2 antagonist/inverse agonist to show that it blocks the effect of swelling on GPCR cascade activation.

Concerning the control with inverse agonists, Indeed, we conducted this experiment (reported in **Supplementary Figure 6** and now also in **Figure R7**). Intriguingly we could observe that blockage with propranolol can be 'broken'-through in swollen cells, further suggesting that swelling favours an active conformation of the receptors, and at least transiently increases the K_{OFF} of the antagonist, allowing the agonist to bind (and NB80 to be recruited / downstream cAMP to be produced). For this reason, we are unsure whether selective blocking of either beta receptors as suggested by the reviewer would help us out in this context. To disentangle the contribution of the two receptors, as mentioned above, we expressed either receptors in a beta-AR free cell line such as CHO.

In terms of general concepts and translational relevance, how does the pathology (e.g. ischemic injury mentioned by authors) would affect their measurements directly? Would other ways of affecting cell/myocytes size such as stretching would have similar effects?

We have not considered uniaxial stretching but have explored applying localized deformations (and force) using optically manipulated microspheres (see below point 3 reply to reviewer #3). In these preliminary experiments we could observe mild receptor conformational activation at the site of deformation, suggesting that a combination of geometrical deformation and associated cortical actin remodelling, mimicking part of what occurs upon swelling, may be at the source of the observed effects.

In terms of translational relevance our observations would suggest that upon reperfusion following osmotic injury, adrenergic response would be enhanced, increasing the adrenergic stimulation in addition to the previously reported catecholamine release upon reperfusion in cardiomyocytes (10.1161/01.CIR.0000128040.43933.D3)

Reviewer #3 (Remarks to the Author):

The study by Sirbu and colleagues investigates the effect of hypoosmotic stress on the signaling of β 2-adrenergic receptors (β 2-AR) in living cells. The authors identify that cell swelling leads to an increase in the isoproterenol-driven cAMP response in HEK293 cells (Fig. 1) and cardiomyocytes (Fig. 2). Increased cAMP levels are also observed after activation of the Gs-coupled histamine and melanocortin receptors, and the response of the Gi-coupled opioid receptors is also enhanced by cell swelling. The authors subsequently perform several experiments aiming to provide mechanistic understanding of the enhanced cAMP transients upon β 2-AR activation and identify a role for Gs in the response to cell swelling (Fig. 3). Using protein relocalization assays, the authors then show an increased recruitment of the Gs activation sensor Nb37 and a faster recruitment of β -arrestin2 to the plasma membrane in hypoosmotic stress conditions (Fig. 4). Finally, the authors present evidence for an increased isoproterenol affinity for β 2-AR and a left-shifted concentration response curve for β 2-AR activation (Nb80 recruitment) in hypotonic conditions (Fig. 5). Based on the results, the authors propose the model that cell swelling favors the active β 2-AR conformation, which would enhance G protein coupling and thereby an increase in cAMP production.

The study addresses an important research question, namely how mechanical forces effect intracellular biochemical signals, which is an emerging topic in GPCR biology and could be relevant to various physiopathological conditions. *However, the experimental data are limited, often indirect, and do not offer a molecular explanation for the effect of cell swelling on GPCR activation and cAMP signaling.* Therefore, the hypothesis that osmotic swelling directly impacts β 2-AR's conformation and ternary complex formation appears speculative based on the data presented in this study. Several key open questions remain and alternative explanations for the observed cAMP response increase need to be ruled out. Furthermore, it is not clear how the cell swelling conditions relate to physiological settings. *In addition, the authors do not acknowledge existing studies on class A GPCRs, which have already investigated the effect of osmotic shock on receptor signaling (see below).*

We thank the reviewer for this overall assessment. Nonetheless, we believe our data are adequate in pointing to a conformational modification at the receptor level, short of any structural investigation that would be impossible to conduct in situ in swollen cells. We address the specific concerns in a point-by-point fashion below.

The major concerns are listed below:
1. Cell swelling/hypotonic solution induces a reduction of clathrin-mediated endocytosis (e.g. PMID: 28923837 & 35389747). The authors should probe how cell swelling affects the trafficking and desensitization process of β 2-AR. It is possible that elevated receptor levels and/or a block in endocytosis are underlying the changed cAMP profile. The authors should carefully measure the effect of cell swelling on cell surface β 2-AR levels, i) before agonist (to determine possible alterations in baseline receptor levels) and ii) upon agonist treatment.

Albeit the timescale of the effects we observe is somehow faster (G protein recruitment) than the typical onset of endocytosis (after beta-arrestin recruitment), we agree with the reviewer that controls for receptor expression and effect of endocytosis would be important. For this reason, we conducted experiments such as those reported in **Figure 3B**, that allowed us observing that the effects of swelling upon isoproterenol-stimulated cAMP production persist also upon the inhibition of endocytosis by Dyngo-4a. In terms of membrane receptor expression, we show now in **Figure R2**, that receptors levels before and after swelling are the

same, and before and after agonist treatment the decrease in receptor at the membrane is the same (down to 70% of the unstimulated receptor) in both control and swollen cells.

2. Cell swelling at 200 mOsm is performed in all presented experiments, however the rationale for the chosen osmolarity and time point are not mentioned. It appears that cell swelling continues linearly with time (Fig. 2b). Are the experimental conditions relevant to (patho-)physiological conditions? How robust is the effect on the cAMP response across hypotonic stress conditions, i.e. how is signaling affected at various osmolarities and at different time points upon switch to swelling media?

We thank the reviewer for this observation. Upon rechecking the source data in **Figure 2B** we identified an outlier. While also in the original data it is clear that the trend is not linear, and that the slope decreases over time (meaning that adult CMs' volume will likely stabilize at 30 minutes), upon removal of the outlier the trend is much clearer (**Figure R10**). On the other hand, for HEK293 it is obvious (thanks also to the larger sample size) how cells volume stabilizes after 15 minutes, as displayed in **Figure R11** below.

All our cAMP readout experiments were conducted in a kinetic fashion, not as endpoint assays. The timepoints used to calculate EC_{50} values were determined by the timepoint at which cAMP traces peaked, e.g. approximately 15 minutes after inducing swelling, and about 5 minutes after agonist stimulation.

Fig. R10 Maximal CM area is reached after ca. 20 minutes of swelling induction average curve showing the area of a confocal CM slice with interrupted line showing the one phase association fit, while dotted lines represent the baseline and maximal area, according to the fit (mean \pm SEM; n = 6 cells from 3 independent experiments)

Fig R11 Quantification of changes in cell volume upon osmotic swelling
a,c confocal slices in XY (**a**) and XZ (**c**) projections in isotonic medium (left), after 10 minutes in hypotonic medium of 200mOsm (center) and an overlay of cell borders (right); **b,d** average area change over time (mean \pm SEM, $n = 33$ cells from 3 independent experiments (**b**) and $n = 7$ cells from 3 experiments (**d**)).

Concerning the determination of the osmolarity for our experiments, and to address the reviewer's question about the effect of varying osmolarities, we could observe that the peak of cAMP response to ISO is proportional to the osmolarity, and reverses upon changing from hypotonic to hypertonic conditions, as clearly illustrated in **Figure R12** below.

Fig. R12 Osmolarity can be "titrated" to differentially modulate cAMP production
a averaged curves of intracellular cAMP concentration under different osmolarities **b** Areas under the curve obtained from single experiments and plotted against osmolarity, including a linear regression fit ($n = 6$ plates from 2 days)

3. Cell swelling results in increased plasma membrane tension (e.g. PMID: 34785592), yet the authors do not provide any experiment / do not discuss the possibility that surface tension affects the activation of β 2-AR/Gs. It is already well established that multiple GPCRs can be activated by mechanical stimuli including cell stretching, shear stress, and hypotonic shock (mentioned here as cell swelling) (e.g. PMID: 31857598, 17030791, 25170081, <https://www.sciencedirect.com/science/article/pii/S2468867323000597#bib48>)

We thank the reviewer for this comment. Indeed, the effect of mechanical tension along the plasma membrane has been invoked as the cause for the activation of selected GPCRs, such as the bradykinin B2 receptor investigated early on in the work of John Frangos (Chachisvilis et al 2006, 10.1073/pnas.0607224103), or the effects reported by Erdogmus et al in 2019 on the Histamine H1R (10.1038/s41467-019-13722-0) A recent survey is also provided in the recent minireview by Hardman et al. from 2023 (10.1016/j.cophys.2023.100689). However, in all these instances we see GPCRs that undergo activation upon pure mechanical stimulation. **This is fundamentally different from our observations, that do not display spontaneous activation upon hypotonic shock at 200 mOsm, but only an effect upon additional agonist stimulation.**

We shall note that the only report that discusses the effect of hypotonic shock also on ligand dependent stimulation of a GPCR, is the 2014 Tang et al Journal of Biological Chemistry (10.1074/jbc.M114.585067) discussing the allosteric effect of hypotonic shock in favouring recruitment of beta-arrestin2 to the AT1R. Moreover, this work uses a receptor-arrestin fusion, that may clearly bias the results, as opposed to our recruitment of a fluorescently labelled arrestin. These data are in excellent agreement with the observations we report in **Figure 4e-h**, again pointing to a potentially more general effect at play.

The known role of membrane tension on GPCRs / β 2-AR must be mentioned and discussed in introduction and discussion. Furthermore, if the authors think that membrane tension does not contribute to the here observed potentiation of the Iso effect, additional experiments need to prove that membrane tension is not affected upon cell swelling (e.g. using Flipper-TR membrane tension probes) and that parallel ways to increase membrane tension (e.g. cell stretching) do not promote a similarly enhanced cAMP response.

We appreciate the suggestion from the reviewer of including a discussion of membrane tension in the activation of GPCRs, keeping in mind the caveats we highlighted in the previous paragraphs. To the point of whether 200 mOsm hypotonic shock causes sufficient membrane stretch to induce sizable tension, the recent work from Roux and collaborators (10.1073/pnas.2103228118), does not appear to show any quantifiable change in membrane tension for Osmolarities within 100 mOsm from isotonic conditions. Specifically, changes in membrane tension begin to show up upon swelling at 120 mOsm. These data support findings from 2018 (10.1073/pnas.2103228118) displaying no change in membrane tension for 240 mOsm conditions.

On the experimental side, we conducted three experiments to address this point:

1. We measured the effect of our swelling conditions on membrane tension using Flipper-TR, as suggested by the reviewer. We shall point out that in the best-case scenario, lifetime changes within just 1 ns are to be expected. Within 10 minutes of 200 mOsm swelling of HEK293 cells, we did observe a relatively small change to the lifetime of Flipper-TR, as measured on a Leica Stellaris confocal microscope. This is here summarised in the form of a Phasor Plot for the two conditions, that shows only a slight change of τ_M (modulation lifetime) and τ_Φ (phase lifetime) upon swelling

(**Figure R13**). These data do not exclude an effect of membrane tension, but indicate that if that is present, is at the limit of what is quantifiable using the available tools.

2. It has recently been shown by De Belly et al (10.1016/j.cell.2023.05.014) that cortical actin integrity is necessary to support membrane tension propagation. This would seem to support the notion that upon swelling and breaking down the integrity of the cortical actin, tension propagation would be impaired, and any mechanism relying on transmission of membrane tension may be affected. However, the enhancement of cAMP response upon cell swelling is unaffected in cells pre-treated with Latrunculin B, as shown in **Figure R14**.
3. We conducted an experiment where we monitored a conformational GPCR biosensor namely the $\beta 2$ -ARcpGFP (kindly provided by Dr. Tommaso Patriarchi). This experiment is conducted in a confocal microscope coupled to a micromanipulation device, i.e. the C-trap (Lumicks). Here, instead of modulating membrane and cortical biophysical properties by hypotonic shock, we deformed the membrane of HEK293 cells incubated in 1 μ M isoproterenol by using 4 μ m polystyrene beads, being pushed against the cell membrane. Albeit preliminary, these data indicate that there is a trend towards a somewhat more active receptor conformation at the site of the deformation, as shown in **Figure R15**. Arguably, this could be caused by a change in membrane tension and/or curvature, having ruled out actin remodelling as a contributing factor in **Figure R14**.

In summary, our measurements seem to indicate that a combination of transient, small membrane tension changes and alteration of cell curvature could play a role in the observed effect, albeit confined to transient and more local effects that are not mediated by the cortical actin cytoskeleton. While we believe that elucidating whether a molecular connection between hypotonic shock (i.e. osmotic swelling) and membrane tension is important, we shall note here that we feel this is beyond the scope of the current manuscript, and should be the focus of a separate, follow-up investigation.

Fig. R13 Phasor plot shows a slight increase in membrane tension in swollen cells
Phasor plots representing the lifetime readout of the membrane tension sensor Flipper-TR sensor in basal conditions (green) (**a**) and after 10 minutes of hypotonic treatment (red) (**b**); **c** overlay of the isotonic (green) and the hypotonic (red) conditions.

Fig. R14 Cytoskeletal integrity is not required to mediate the observed effect.

Representative kinetic curve of acceptor/donor ratio measured in HEK293T cells stably expressing Epac-S^{H187} in a microplate reader, while treated with 10 μM Latrunculin B – an actin depolymerizing agent (N=1).

Fig. R15 Effect of applying a membrane deformation to HEK293 cell expressing the β2-ARcpGFP conformational biosensor

A 4 μm polystyrene bead is sequentially approached to the membrane of a HEK293 cell, perfused with 10 μm isoproterenol. The fluorescence intensity of the cpGFP biosensor is recorded from two areas, at the back of the cell and from a portion of membrane facing the approaching bead. As contact is established (frame 9) we observe a slight increase in the fluorescence intensity of the region being deformed by the bead (blue trace in the chart).

4. The authors show enhanced AC activity in cell swelling conditions independently of β2-AR activation, which is dependent on Gs (Fig. 3d). How do the authors explain this result? Does cell swelling drive an elevated basal Gs tone? What is the mechanism and how does it influence the β2-AR pathway studied in the other experiments?

We thank the reviewer for this comment, which raises an important point. As highlighted in response to reviewer #1, point 5, FSK and Gs have a synergistic effect, and in the absence of Gs, cAMP response to FSK is blunted or abolished altogether (10.1016/j.neuropharm.2005.11.004, science.278.5345.1907, 10.1074/jbc.272.35.22265). Indeed, if we had observed spontaneous receptor activation upon swelling, we could ascribe enhanced response to FSK in swollen cells to increased number of active Gαs subunits. However, since we do not observe cAMP increase upon osmotic swelling -unless isoproterenol is added- we believe that the enhanced response to FSK that we report in **Figure 3d** has an independent origin: it may either indicate a sensitization of AC catalytic activity or be ascribed to an enhanced permeability of the stretched plasma membrane to FSK. Either way, we believe that these observations do not affect in the least our conclusions concerning

the beta2-AR, in particular as they are supported by the receptor-specific data reported in **Figures 4 and 5** of our manuscript.

5. The observed increase in transducer protein recruitment upon cell swelling is based on TIRF experiments, which measure relocalization of proteins from the cytosol to the plasma membrane in close contact with the glass cover slip. Given that cell swelling leads to a flattening of the membrane (described by the authors as ‘ironing out wrinkles in a cloth’ in the introduction), the TIRF signal may directly be affected due to changes in membrane topology and may underlie the detected transducer protein signal increase. The authors need to control for such a possibility by including intensity traces of PM localized or PM-recruited control proteins that do not show an altered signal.

We thank the reviewer for this suggestion, and we have indeed performed the required controls. We used a membrane-bound fluorescent protein construct, namely Venus-CAAX (**Figure R16a**), to monitor potential increases of fluorescence, that would be indicative of a geometric effect in enhancing membrane fluorescence of recruited proteins upon swelling. Any effect that we could see is confined to within $\pm 3\%$ of the baseline (**Figure R16b**), suggesting that geometric effects causing more fluorescent signal coming to the TIR field do not clearly play any significant role.

Fig. R16 Cell swelling does not change fluorescence intensity of a membrane anchored fluorescent protein under TIRF-M measurements.

a representative TIRF-M images of basolateral membrane of HEK293AD cell transiently transfected with Venus-CAAX before (left) and after (right) medium change for the control (upper) and swelling (lower) condition; **b** averaged curves of relative fluorescence intensity change upon medium replacement (mean \pm SEM; $n = 10$ cells from 2 independent experiments)

6. The observed change in arrestin recruitment upon cell swelling does not seem to be an increase in recruitment but rather a faster recruitment (Fig. 4g). How do the authors explain that G protein activation is increased (Fig. 4c) and arrestin recruitment accelerated? Is cell swelling affecting the diffusion of transducer proteins differently? The authors mention that protein diffusion is not modified in the cell swelling condition, yet the data is not included in the manuscript. The data should be added.

We thank the reviewer for this insightful comment. The data displayed in **Figure 4g-h** display both, an increased recruitment (normalized to a saturating 10 μ M ISO concentration) as well as, undoubtedly, a faster kinetics of recruitment. If the receptor is ‘biased’ by swelling towards

an active conformation, this shall induce a higher turnover of G protein at the receptor, which is indeed confirmed by our Nb37 data in **Figure 4a-d** and Nb80 data in **Figure 5e-f**. We then face two possibilities: 1. The conformation assumed by the receptor is also biased towards beta-arrestin, favouring a more rapid and enhanced recruitment or 2. Enhanced beta-arrestin recruitment at the receptor follows closely enhanced G protein recruitment and activation. As the recruitment of beta-arrestin2 is similarly enhanced by swelling in $G\alpha_s$ -KO cells (**Fig. R18** under point 2 of the reviewer 4 response), we are inclined towards the first explanation. Concerning diffusion coefficients, this was indeed our first hypothesis. We had measured diffusion rates of beta receptors as well as of G proteins but could observe only a slight increase of beta2-AR diffusion rate upon swelling and no change in $G\alpha_s$ diffusion speed. (**Figure R17**).

Fig. R17 Single molecule tracking shows faster β 2AR plasma membrane diffusion in swollen cells

a averaged mean squared displacement curves obtained by tracking Snap- β 2AR diffusing on basolateral membrane of transiently transfected HEK293AD cells using TIRF-M; **b** diffusion coefficients calculated per cell from the first 0.15 s of corresponding MSD plots ($MSD = 4Dt$) (mean \pm SEM; $n =$ at least 14 cells from 3 independent experiments; statistical analysis by an unpaired t-test); **c** averaged mean squared displacement curves obtained by tracking $G\alpha_s$ -eYFP diffusing on basolateral membrane of transiently transfected HEK293AD cells using confocal microscopy-based linescan-FCS (mean \pm SEM; $n =$ at least 7 cells from 2 independent experiments).

7. It is a tempting hypothesis that increased membrane tension favours an intermediate state closer to the active GPCR conformation, with increased affinity of β 2-AR for Iso and decreased affinity for an antagonist, which is briefly mentioned in the discussion, with data presented in Supp. Fig 6. Why do the authors 'hide' the data in the supplementary material?

We thank the reviewer for this observation. We would be happy to move our antagonist data, together with the new measurements shown in **Figure R7** to the main Figures.

Reviewer #4 (Remarks to the Author):

Sirbu et al. demonstrate that cellular osmotic swelling potentiates β 2AR ligand-mediated acute cytosolic and plasma membrane cAMP generation in HEK293 cells, as well as cytosolic cAMP in adult mouse ventricular cardiomyocytes. This swelling-mediated effect was also observed upon stimulation with ligands selective for distinct Gs and Gi-coupled endogenous receptors in HEK 293 cells. Additionally, using the HEK 293 cell model, the authors demonstrate that the increased ligand potency for cytosolic cAMP generation with osmotic swelling persists with inhibition of either receptor internalization or phosphodiesterase activity, and with direct stimulation of adenylyl cyclase. However, Gs protein knockdown prevented the osmotic swelling-induced cAMP potentiation with adenylyl cyclase activation. This was consistent with increased Nb37-mYFP recruitment to membrane upon β AR stimulation and osmotic swelling, suggesting that swelling increases Gs-receptor association. Similar results are observed for β -arrestin recruitment. Through fluorescent ligand displacement experiments and Nb80 recruitment assays, the authors show that swelling increases receptor-ligand affinity and promotes an active receptor conformation.

While the concept of osmotic swelling-induced changes in receptor allostery is not new (for instance, see PMID 25170081), this is the first such study to quantify osmotic swelling-induced changes in ligand potency on cAMP generation in response to β 2AR stimulation.

We thank the reviewer for his supportive comments, and we apologise for not having given sufficient emphasis in our introduction to the past work of Tang et al, albeit this was specific to beta-Arrestin downstream of AT1R. This will be certainly included in the revised version of the manuscript.

Comments

1. While this is a well-designed molecular pharmacology study characterizing the phenotype of the potentiation of cAMP signaling, it does not offer biophysical/mechanistic insight into what structural changes are induced by osmotic stress or domains/amino acid residues of β 2AR required for the effect to which water molecules presumably bind.

We thank the reviewer for this insightful comment. We felt that, upon displaying Nb37/80 recruitment data, we had provided a significant degree of biophysical data to address the mechanism at play. We felt that a mutagenesis study of the beta2-AR residues potentially involved in mediating the allosteric network due to cell swelling would be beyond the scope of our study. Moreover, we tried to work as often as possible with endogenous beta receptors, meaning that any potential mutant would require a Crispr/Cas9 clone study to appraise its effect.

However, we conducted a set of classical experiments to further investigate the role of the actin cytoskeleton and membrane lipid composition in modulating the observed effects.

First, we depolymerized actin by the well-established use of Latrunculin B. Actin depolymerization is known to affect also tension propagation along the plasma membrane. **Figure R14** displays a clear peak in cAMP response to ISO also in cells subjected to pre-incubation with 10 μ M Latrunculin B. Based on current reports that membranes of cells where actin has been depolymerized do not support tension propagation, this experiment seems to suggest that the observed effect is not predominantly mediated by increased membrane tension affecting the receptors, but rather by a more complex interplay of processes. This appears to be supported also by the data of our membrane tension measurements of **Figure R13**.

2. Most of the study focuses on the osmotic swelling-induced increase in ligand potency of the Gs-cAMP effect, but Fig4f-h indicates enhanced β -arrestin recruitment as well. Since ternary complexes containing β 2AR, G protein (s or i) and β -arrestins have been shown, is this effect also dependent on G protein expression?

We thank the reviewer for this insightful suggestion. We have conducted now a measurement of beta-arrestin recruitment in Gs KO cells, as displayed below in **Figure R18**. As it can be seen, even in the absence of Gs recruitment of beta-arrestin is sizable, albeit somewhat smaller than what is observed in WT cells. Swelling leads to a large enhancement in beta-arrestin recruitment. These findings appear to suggest that the conformation achieved by the receptor upon swelling is conducive, to both an enhanced Gs as well as beta-arrestin recruitment, with beta-arrestin recruitment more independent of Gs in swollen than in control cells.

Fig. R18 Swelling leads to an increased β -arr2 recruitment independent of $G\alpha_s$

a representative curve showing relative increase of membrane fluorescence collected as first 33 nM, and then at saturating 10 μ M concentration of isoproterenol is added to the cell (normalized to baseline and 10 μ M Iso response); **b** relative fluorescence increase measured upon β -Arr-2-eYFP recruitment upon 33 nM isoproterenol stimulation in swollen vs non-swollen cells (mean \pm SEM; n = 21 (control) and 1 (swelling) cells from 3 independent experiments; statistical analysis performed by an unpaired two-tailed t-test).

3. The authors show osmotic stress also increases cAMP generation in response to the same level of ligand stimulation in adult cardiomyocytes. There is substantial literature describing compartmentation of β AR signaling in cardiomyocytes that relays distinct functional outcomes, does osmotic stress alter this compartmentation? The authors suggest these effects could relate to how receptors in cardiomyocytes may respond to ischemic injury but have not followed through to determine if there is a functional effect on the cardiomyocytes either by osmotic stress alone or in combination with ischemic stress.

We thank very much the reviewer for their comments and suggestions.

There is a consensus that upon osmotic swelling, e.g. upon reperfusion, substantial alterations of ventricular cardiomyocyte function take place, in particular alterations of excitation contraction coupling, with a loss of systolic and diastolic function underpinned by dysregulation of calcium homeostasis (10.1152/ajpheart.2000.279.4.H1963).

Based on the literature available, the physiological role specifically played by adrenergic signalling and downstream cAMP in this context is less clear. Early reports from Hilal-Dandan

and Brunton (10.1152/ajpheart.1995.269.3.H798) suggested an actual decrease of cAMP upon hypoosmotic shock, speculating that upon swelling we may be observing a mechanoactivation of the inhibitory Gi. In contrast, in S49 cells, the opposite effect, namely an increase of cAMP, was observed by the authors.

It has been speculated that the physiological implications of adrenergic stimulation in the context of osmotic swelling could be regulatory volume decrease (10.1085/jgp.110.1.73) by activation of chloride channels.

With respect to these earlier data, our measurements offer now a temporally resolved, and high sensitivity description of isoproterenol-induced cAMP transients upon cell swelling: the fact that only a transient effect is observed, appears to support the notion that this is a regulatory reaction aimed at maintaining cell homeostasis.

Investigating the impact of swelling in terms of cardiomyocytes compartmentalization would be very challenging, in terms of generating several models with targeted cAMP sensors in different localization of the cell. Nonetheless, we provided evidence (**Supplementary Figure 3c**) that in HEK293 cells there are no differences between the cAMP enhanced response in swollen cells near the plasma membrane and that observed in the cytosol, providing at least a hint that this specific response we observe may not be strictly compartmentalized.

4. Dyngo-4 treatment on its own reduced both the potency and efficacy of ISO-mediated cAMP generation; does this suggest that the cytosolic cAMP generation being measured is primarily due to endosomal receptor activation that is altered by Dyngo-disrupted receptor trafficking, and does the osmotic swelling correct for that independent of a direct structural change in the receptor?

We thank very much the reviewer for raising this interesting observation. The fact that a significant portion of cAMP signalling arises from receptors in the endosomes is not new, and we believe our measurements are in line with earlier observations (as reviewed in 10.1016/bs.pmbts.2015.03.008). We interpret our data (**Supplementary Figure 3a**) as indicating that, for those receptors remaining at the plasma membrane outwit clathrin coated pits -yielding the cAMP response-, swelling consistently induces a conformation favouring ligand binding and downstream coupling of the signalling partners.

REVIEWER COMMENTS

Reviewer #1 (Remarks to the Author):

The authors have done extensive revision to address all concerns raised in the previous review. I have no further question.

Reviewer #2 (Remarks to the Author):

The authors have addressed my comments satisfactorily

Reviewer #3 (Remarks to the Author):

The revised manuscript addresses several major concerns I have raised by providing new data and more thorough explanations / discussions. Overall, the revised manuscript is improved. The data that show an elevated cAMP response upon β 2AR activation by Iso in osmotically swollen cells relative to control cells are convincing. Regarding the possible underlying mechanism, I am skeptical that the experiments presented convincingly demonstrate that cell swelling enhances 'GPCR ternary complex formation' (as stated in the title). The conclusions are based on experiments with sensors (Nb80, Nb37), and do not provide a direct biochemical confirmation for an increase in Iso- β 2AR-Gs ternary complexes. Therefore I think the title should probably be adapted / toned down.

Other comments:

- Related to Fig. 4 & Supp. Fig. 9: Have the authors tested a difference in the diffusion of beta-arrestin? Beta-arrestin shows very rapid recruitment in osmotically swollen cells and thus testing arrestin diffusion would be relevant (likely more so than Gs, since Gs shows increased but not accelerated recruitment to β 2AR in swollen cells)
- Fig. R16 is an important control and should be included as supplementary figure.

Minor:

- It is not clear to me why the Nb37 data in Figure 4a-d and Nb80 data in Figure 5e-f confirms 'a higher turnover of G protein at the receptor' (see point-by-point response by authors). Can the authors clarify this statement?

Reviewer #4 (Remarks to the Author):

The authors have adequately addressed my most concerns, in part with new data. It is recognized that an extensive campaign of receptor mutagenesis and subcellular compartmentation experiments to comprehensively define the effects of osmotic swelling on enhanced β 2AR signaling is beyond the scope of this present study.

However, the major question that remains is whether osmotic swelling in the adult cardiomyocytes relays a measurable physiologic response, for instance an increase in the ISO-mediated contractile response.

A second, less crucial point, is that since osmotic swelling-induced β arr recruitment is still enhanced in response to ISO in GsKO cells, the authors may consider whether this is a Gi protein-dependent mechanism, since Gi-dependent β arr recruitment to β ARs has been demonstrated in the literature.

Reviewer #1 (Remarks to the Author):

The authors have done extensive revision to address all concerns raised in the previous review. I have no further question.

Reviewer #2 (Remarks to the Author):

The authors have addressed my comments satisfactorily

Reviewer #3 (Remarks to the Author):

The revised manuscript addresses several major concerns I have raised by providing new data and more thorough explanations / discussions. Overall, the revised manuscript is improved. The data that show an elevated cAMP response upon b2AR activation by Iso in osmotically swollen cells relative to control cells are convincing. Regarding the possible underlying mechanism, I am skeptical that the experiments presented convincingly demonstrate that cell swelling enhances 'GPCR ternary complex formation' (as stated in the title). The conclusions are based on experiments with sensors (Nb80, Nb37), and do not provide a direct biochemical confirmation for an increase in Iso-b2AR-Gs ternary complexes. Therefore I think the title should probably be adapted / toned down.

We thank the reviewer for their comments and for the positive feedback on the revised manuscript. We propose here a new manuscript title, **Cell swelling allosterically enhances β -adrenergic signaling**, de-emphasizing the aspect about ternary complex formation that the reviewer points out.

Other comments:

- Related to Fig. 4 & Supp. Fig. 9: Have the authors tested a difference in the diffusion of beta-arrestin? Beta-arrestin shows very rapid recruitment in osmotically swollen cells and thus testing arrestin diffusion would be relevant (likely more so than Gs, since Gs shows increased but not accelerated recruitment to b2AR in swollen cells)

We did not test arrestin diffusion, as it is not a constitutively membrane-anchored protein: for this reason single-molecule TIRF microscopy is almost impossible and also Fluorescence Correlation Spectroscopy-based approaches (FCS) are very challenging as the measured diffusion would unavoidably include also a cytosolic component that would mask any readout at the membrane. Alternatively, the rate of basal arrestin recruitment to the membrane (unspecific lipid anchoring) could be compared before and after swelling using e.g. bystander BRET or NanoBit assay, but we believe this would fall within the scope of further studies.

- Fig. R16 is an important control and should be included as supplementary

figure. We have included Figure R16 in the manuscript as **Supplementary Figure 8**

Minor:

- It is not clear to me why the Nb37 data in Figure 4a-d and Nb80 data in Figure 5e-f confirms 'a higher turnover of G protein at the receptor' (see point-by-point response by authors). Can the

authors clarify this statement?

Although we indeed do not measure G protein turnover directly, the nanobody experiments serve as proxies for the first two steps of G protein activation: G protein recruitment to the active receptor (Nb80) and GDP release (Nb37). We indeed do not monitor GTP binding and G protein dissociation.

Reviewer #4 (Remarks to the Author):

The authors have adequately addressed my most concerns, in part with new data. It is recognized that an extensive campaign of receptor mutagenesis and subcellular compartmentation experiments to comprehensively define the effects of osmotic swelling on enhanced β 2AR signaling is beyond the scope of this present study.

However, the major question that remains is whether osmotic swelling in the adult cardiomyocytes relays a measurable physiologic response, for instance an increase in the ISO-mediated contractile response.

We thank the reviewer for their remarks and share their interest in how the observed effects may translate to functional responses. Unfortunately, we currently do not have access to primary adult cardiomyocytes to conduct contractile response measurements in the same setting as the cAMP measurements. Furthermore, the cAMP experiments we conducted earlier were done under contraction inhibition by para-Aminoblebbistatin (a myosin II inhibitor) and no contractile data can be extracted from past experiments.

Nevertheless, we have conducted a set of experiments using hiPSC-derived cardiomyocytes, which upon approximately 60 days of differentiation had reached a mature, beating stage, following a strategy we adopted earlier: crucially, we could show that at this timepoint of differentiation, the cells expressed **both β -ARs** (Gmach et al. IJMS 2022).

In our new experiments we observe an increase in beating frequency for the cells stimulated by the same concentration of isoproterenol in hypoosmotic medium (200 mOsm), compared to isoosmotic medium (300 mOsm). This demonstrates the functional relevance of increased cAMP concentrations observed by us in HEK293 cells and primary cardiomyocytes. We have included these data as **Supplementary Figure 5** in the manuscript, and include them also here.

Supp. Fig. 5: Characterization of hiPSC-CM chronotropic response upon swelling **a** Schematic of the experimental setup. **b** Three wells of the same batch of hiPSC-CM were imaged in a brightfield microscope first in isosmotic (300 mOsm) and then hypotonic (200 mOsm) conditions upon addition of 100 nM iso. Kymographs were collected from line selections to determine the beating rate. **c.** Beating rate before and after swelling for the three sets of experiments.

A second, less crucial point, is that since osmotic swelling-induced 13arr recruitment is still enhanced in response to ISO in GsKO cells, the authors may consider whether this is a Gi protein-dependent mechanism, since Gi-dependent 13arr recruitment to 13ARs has been demonstrated in the literature.

We thank the reviewer very much for their comments about Gi signaling. We had indeed conducted a pilot experiment in this direction using a Gi FRET biosensor, and indeed it appears that there is an increased coupling to Gi as well (Figure R19), at least for the β 1-AR. At this point, incorporating a complete set of data on Gi activation appears beyond the current scope of the manuscript, but we believe this could be an exciting direction for follow up work.

Fig. R19 Enhanced G_i activation (monitored by the dissociation of a FRET biosensor) in response to isoproterenol in swollen cells for the $\beta 1$ -AR and $\beta 2$ -AR

a,c concentration-response curves and **b,d** representative Δ FRET curves comparing G_{i2} FRET-sensor response to isoproterenol in control and swollen HEK293 cells, transiently transfected with $\beta 1$ -AR or $\beta 2$ -AR and the G_{i2} -FRET sensor

REVIEWERS' COMMENTS

Reviewer #3 (Remarks to the Author):

I thank the authors for responding to my comments, I am now in favour of publication.

Yet, I recommend that the authors reassess the new title 'Cell swelling allosterically enhances β -adrenergic signaling' since the word 'allostery' does not accurately reflect the key message of the paper in my opinion. In addition, it is not quite clear how cell swelling would act 'allosterically'.

Possible alternatives could be 'Cell swelling enhances canonical β -adrenergic signaling' or 'Cell swelling enhances / potentiates ligand-driven β -adrenergic signaling'.

Reviewer #4 (Remarks to the Author):

The authors have adequately addressed my concerns, in particular the addition of new experiments using hiPSC-derived cardiomyocytes to demonstrate physiologic relevance of osmotic swelling on enhanced betaAR signaling (enhanced beat frequency).

The data included in the response to reviewers regarding the possible role for Gi-dependent signaling in the hypo-osmotic effect is indeed exciting; this reviewer will look forward to reading a follow-up study by the authors on this aspect.

St Andrews August 9th 2024

Reviewer #3 (Remarks to the Author)

I thank the authors for responding to my comments, I am now in favour of publication.

Yet, I recommend that the authors reassess the new title 'Cell swelling allosterically enhances β -adrenergic signaling' since the word 'allostery' does not accurately reflect the key message of the paper in my opinion. In addition, it is not quite clear how cell swelling would act 'allosterically'.

Possible alternatives could be 'Cell swelling enhances canonical β -adrenergic signaling' or 'Cell swelling enhances / potentiates ligand-driven β -adrenergic signaling'.

We have modified the title as recommended by the reviewer as 'Cell swelling enhances ligand-driven β -adrenergic signaling'

Reviewer #4 (Remarks to the Author)

The authors have adequately addressed my concerns, in particular the addition of new experiments using hiPSC-derived cardiomyocytes to demonstrate physiologic relevance of osmotic swelling on enhanced betaAR signaling (enhanced beat frequency).

The data included in the response to reviewers regarding the possible role for Gi-dependent signaling in the hypo-osmotic effect is indeed exciting; this reviewer will look forward to reading a follow-up study by the authors on this aspect.

We thank the reviewer for their final kind comments.